# She1 affects dynein through direct interactions with the microtubule and the dynein microtubule-binding domain

Kari H. Ecklund[1], Tatsuya Morisaki[1], Lindsay G. Lammers[1], Matthew G. Marzo[1], Timothy J. Stasevich[1] & Steven M. Markus[1]

Cytoplasmic dynein is an enormous minus end-directed microtubule motor. Rather than existing as bare tracks, microtubules are bound by numerous microtubule-associated proteins (MAPs) that have the capacity to affect various cellular functions, including motor-mediated transport. One such MAP is She1, a dynein effector that polarizes dynein-mediated spindle movements in budding yeast. Here, we characterize the molecular basis by which She1 affects dynein, providing the first such insight into which a MAP can modulate motor motility. We find that She1 affects the ATPase rate, microtubule-binding affinity, and stepping behavior of dynein, and that microtubule binding by She1 is required for its effects on dynein motility. Moreover, we find that She1 directly contacts the microtubule-binding domain of dynein, and that their interaction is sensitive to the nucleotide-bound state of the motor. Our data support a model in which simultaneous interactions between the microtubule and dynein enables She1 to directly affect dynein motility.

---

[1] Department of Biochemistry and Molecular Biology, Colorado State University, Fort Collins, CO 80523, USA. Correspondence and requests for materials should be addressed to S.M.M. (email: steven.markus@colostate.edu)

The large size and crowded environment of a typical eukaryotic cell necessitates the tightly regulated active transport of myriad cargoes to various subcellular sites. In eukaryotic cells, this transport is mediated by a large family of molecular motors that walk along polarized actin and microtubule filaments (reviewed in ref. [1]). The kinesin and dynein families of microtubule motors are responsible for cargo transport toward the plus and minus ends of microtubules (with few exceptions) that are generally situated at the cell periphery and cell center, respectively. Given the strict spatial and temporal requirements for motor-mediated cargo transport, precisely tuned motor activity is imperative for the development and maintenance of a healthy cell and tissue.

Rather than existing as bare tracks, microtubules are bound by various classes of microtubule-associated proteins (MAPs), including those that bind along the lattice (e.g., MAP1A, tau, TPX2, PRC1[2–5]), those that concentrate at the plus (e.g., EB1, CLIP170, CLASP[6–8]) or minus ends (e.g., Patronin[9]), and a large number of microtubule motors. Several studies have revealed the response of some motors to such "roadblocks". For instance, in vitro studies have revealed that single molecules of kinesin slow down and are more likely to dissociate when encountering either high concentrations of other kinesins[10, 11] or tau[12]. Similar studies have revealed that upon encountering tau, dynein motors tend to reverse direction rather than detach[12, 13], whereas MAP4 (a non-neuronal tau family member) reduces the velocity of dynein motors in vitro[14] and their run length in vivo[15].

In addition to exhibiting "roadblock" activity (i.e., inducing detachment or reducing velocity), several MAPs have been shown to recruit kinesins to various microtubule structures. For instance, studies in several model systems have shown that the microtubule cross-linking protein PRC1 (Ase1 in fission and budding yeasts) is important for the recruitment of the kinesins Xklp1 (Xenopus laevis), Cin8 (budding yeast), and Klp9 (fission yeast), all of which affect spindle midzone functions[16–18]. Similarly, the MAP TPX2 has been shown to be important for recruitment of the kinesin-5, Eg5, to spindle microtubules where it functions in spindle assembly[19, 20]. In addition to a recruitment role, Tpx2 has also been shown to reduce the velocity of Eg5[19, 21]. Thus, understanding how various motors navigate around or are affected by MAPs is critical to understanding the molecular regulation of cellular motor activity.

In contrast to the kinesin family of motors, which are represented by at least 45 proteins in human cells[22], only one variant of cyoplasmic dynein (dynein-1) is encoded by eukaryotic genomes and is responsible for nearly all minus end-directed microtubule transport. Given its varied cellular roles it is unsurprising that numerous regulators contribute to in vivo dynein function. These include LIS1 (human homolog of yeast Pac1), the dynactin complex, and the growing family of adapter proteins that link dynein to dynactin and various cellular sites (e.g., Bicaudal-D, Hook, Spindly). These effectors each exhibit unique activities and mechanisms of action. For instance, the LIS1 homolog Pac1 reduces dynein velocity through direct binding to the AAA (ATPase associated with various cellular activities) ring, which sterically blocks its mechanochemical cycle[23]. The dynactin complex, on the other hand, activates metazoan dynein motility[24, 25] through a mechanism that likely involves promotion of microtubule binding[26], and orienting the two motor domains appropriately for processive motility[27, 28]. These dynactin-mediated activities require adapter proteins that promote binding between dynactin and the N-terminal tail domain of dynein (or tail-bound accessory chains)[24, 25, 28, 29]. Although it is unclear if other regions of the dynein motor are targets for regulation, the size, architecture, and complex mechanochemical cycle of dynein suggest at least the potential for various sites of regulation. For instance, the crowded microenvironment of the microtubule lattice raises the possibility that MAPs may regulate dynein activity via direct interactions with regions of the motor that are in close proximity to the microtubule (i.e., the microtubule-binding domain, MTBD, or the coiled coil (CC) that links the MTBD to the AAA ring). However, no such activity has yet been identified.

Here, we focus on understanding the mechanism by which the MAP She1 affects dynein motility. The role for She1 in dynein function is currently unclear, although in vivo studies have shown that deletion of She1 leads to defects in daughter cell-directed spindle movements, while in vitro studies have shown that She1 is a potent effector of dynein motility[30]. Specifically, She1 reduces dynein velocity and increases the duration of time dynein spends bound to microtubules. Interestingly, She1 exhibits high specificity for dynein and has no apparent effect on the motility of either human kinesin-1 or the yeast kinesin Kip2. Thus, in spite of them possessing distinct cellular roles, She1 and Pac1 (the latter of which is important for plus end-binding activity of dynein[31]) affect dynein motility similarly[32, 33], raising the possibility that She1 affects dynein activity in a similar manner. Using recombinant proteins we show that She1 in fact affects dynein motility using a unique mechanism of action. Through direct binding between the microtubule and the dynein MTBD, She1 reduces dynein microtubule dissociation, which results in reduced ATPase activity, stepping frequency and velocity, and increased microtubule dwell times. We confirm the She1–MTBD interaction by generating a chimeric dynein mutant that exhibits a reduced binding affinity for She1 and is less sensitive to She1 effects in vitro and in vivo. Interestingly, we find that She1 recognizes a specific conformational state of the MTBD that is representative of the nucleotide-free, high microtubule-binding affinity state. Taken together, our findings reveal the first mechanism by which a MAP may affect dynein activity and also reveal the MTBD as a novel target for dynein regulation.

## Results

**She1 reduces dynein ATPase activity**. To understand the molecular mechanism by which She1 affects dynein motility, we first asked whether She1 has any effect on dynein's mechanochemical cycle. It is fairly well established that for every step it takes, dynein binds and hydrolyzes at least one ATP at an active site within the first AAA module (AAA1)[34]. ATP binding and hydrolysis have been shown to trigger a cascade of conformational changes that ultimately lead to (1) movement of the mechanical linker element to its pre-powerstroke state[35–39], and (2) reduced affinity of the dynein MTBD for microtubules[40–42]. Phosphate release (ADP-$P_i$ to ADP) on the other hand is thought to be triggered upon microtubule rebinding[43], which consequently leads to (1) powerstroke of the linker[36, 41, 44], and (2) adoption of a high microtubule-binding affinity conformation of the MTBD[34, 40, 42, 45]. Thus, the ATPase cycle is tightly coordinated with the microtubule-bound state of the motor. One potential mechanism by which She1 may affect dynein motility is through direct modulation of dynein's ATPase activity.

To determine what effect, if any, She1 has on dynein's ATPase activity, we measured the rate of ATP hydrolysis of dynein in response to 0–2 μM microtubules, and in the absence or presence of recombinant She1. For these studies we used a purified, artificially dimerized (via glutathione S-transferase, GST), motility-competent dynein motor domain fragment[46] that is sensitive to She1-mediated inhibition[30] (GST–dynein$_{331}$; see Fig. 1a). We found that She1 indeed reduces dynein's maximal microtubule-stimulated ATPase activity ($k_{cat}$) from $17.7 \pm 1.1$ (SE of fit) to $11.0 \pm 0.4$ motor domain$^{-1}$ sec$^{-1}$ (Fig. 1b, c). However, She1 had no significant effect on the basal ATPase rate (from $2.4 \pm 0.9$ to $3.1 \pm 0.5$ motor domain$^{-1}$ sec$^{-1}$; Fig. 1c, $k_{basal}$), suggesting

that She1 does not directly affect ATP turnover in the absence of microtubule binding. Interestingly, we found that She1 increased the binding affinity of dynein for microtubules, as was apparent by the 2.3-fold reduction in $K_{m(MT)}$ (from $0.10 \pm 0.03$ to $0.04 \pm 0.01$ μM; ± SE of fit; Fig. 1c, $K_{m(MT)}$). These data suggest that She1 may affect dynein motility by directly affecting ATP turnover at one of the AAA modules within the motor domain. Alternatively, given that She1 reduces dynein velocity, it is equally plausible that She1 reduces the rate at which dynein binds and hydrolyzes ATP as a consequence of a reduced stepping rate.

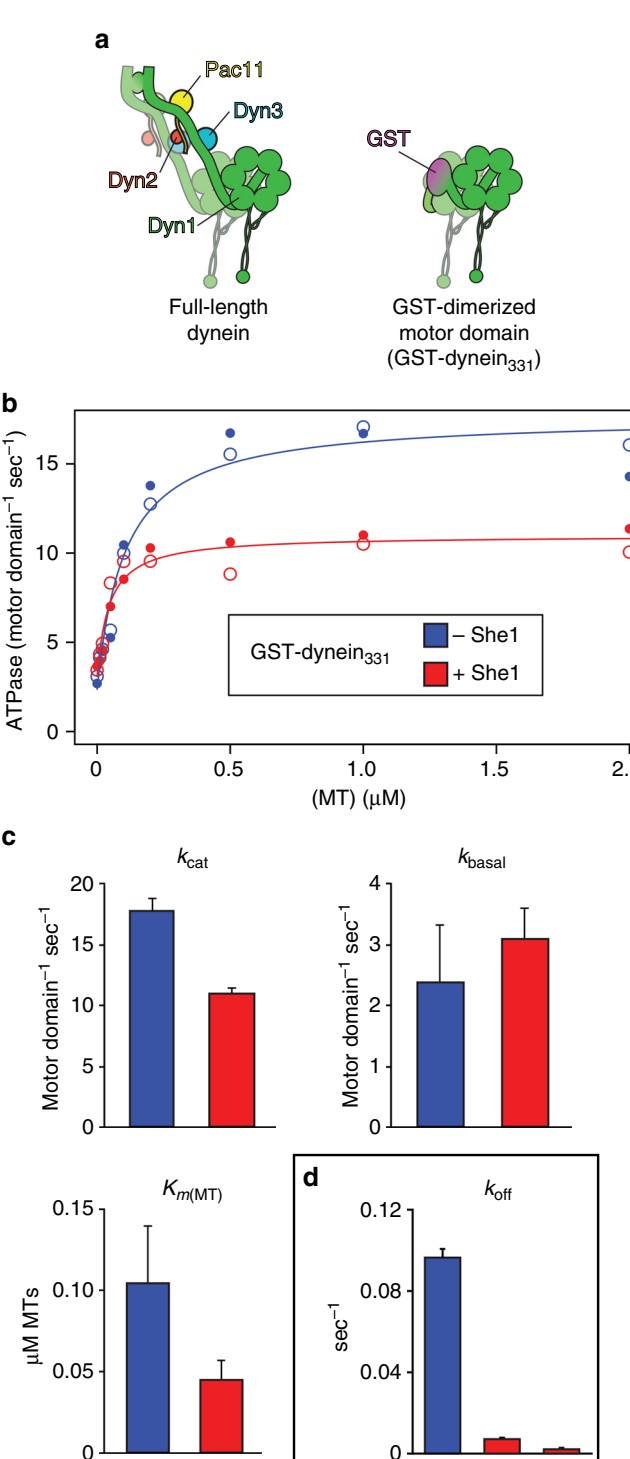

**b** ... **c** ...

**She1 reduces dynein stepping frequency.** The fact that She1 increases the binding affinity of dynein for microtubules (see Fig. 1c, $K_{m(MT)}$), likely as a consequence of reduced dissociation rates (Fig. 1d; as determined from single-molecule experiments), suggests that She1 may slow down dynein motility by prolonging the periods of microtubule attachment between individual steps, thus reducing the overall stepping frequency of dynein. To determine how She1 affects the stepping behavior of dynein, we used an established method[46] to attach a bright photostable quantum dot (Qdot) to the C-terminus of one of the two dynein motor domains within a GST-dimerized complex (Fig. 2a). We then imaged these molecules at high temporal resolution ($\sim$10 sec$^{-1}$) in either the absence or presence of She1. Consistent with previous findings, in the absence of She1 and the presence of saturating ATP concentrations (1 mM), dynein motors moved at a rate that matched or exceeded the temporal resolution of our imaging conditions (Fig. 2b, green trace). This made it difficult to accurately track these motors and thus determine dynein's stepping behavior (e.g., stepping frequency and step size). Thus, we reduced the velocity of dynein by using limiting concentrations of ATP (1 μM), which permitted accurate assessment of dynein stepping behavior due to the longer dwells between individual steps (Fig. 2b, blue traces).

In the absence of She1, the distribution of dynein step sizes revealed a major peak at approximately 16 nm (Fig. 2c) and a small fraction (10.9%) of backwards steps (Fig. 2g), both of which are consistent with previous findings[46]. In contrast to dynein motility in the absence of She1, the presence of She1 sufficiently reduced the stepping frequency of dynein in saturating ATP concentrations (1 mM) to permit the observation of discrete steps with pauses in between (Fig. 2b, red and brown traces). We observed a stepping rate of 2.4 sec$^{-1}$ in the presence of 10 nM She1, which was reduced to 1.2 sec$^{-1}$ by 25 nM She1, a value that closely matched that of dynein alone in 1 μM ATP (1.0 sec$^{-1}$; Fig. 2f and Supplementary Fig. 1c). Interestingly, we also observed an increased fraction of backwards (plus end-directed) and large steps in the presence of She1 (Fig. 2c–e, yellow boxes and Fig. 2g, h). Taken together, our findings reveal that She1 indeed reduces dynein stepping frequency, likely as a consequence of the reduced microtubule dissociation rate (Fig. 1d).

**She1 requires microtubule binding to affect dynein motility.** Given that She1 binds microtubules with nanomolar affinity[30], we next asked whether this activity of She1 is required for it to affect dynein motility. To this end, we proteolytically removed the unstructured carboxy-terminal tails of α-tubulin and β-tubulin

**Fig. 1** She1 reduces dynein microtubule-stimulated ATPase activity, and enhances the affinity of dynein for microtubules. **a** Cartoon representation of the full-length dynein complex (left, with associated accessory chains; Dyn2, dynein light chain; Dyn3, dynein light-intermediate chain; Pac11, dynein intermediate chain; Dyn1, dynein heavy chain), and the minimal GST-dimerized dynein motor domain (right). **b, c** Microtubule-stimulated ATPase activity in the absence (blue) and presence (red) of 200 nM She1. Data points from two replicate experiments are shown (open and closed circles). Data were fit as described in Methods to obtain the basal ($k_{basal}$; microtubule-unstimulated ATPase activity) and maximal ($k_{cat}$; microtubule-stimulated) ATPase rates, and the microtubule concentration at which half-maximal ATPase activation is achieved ($K_{m(MT)}$), all of which are depicted in **c** (error bars, standard error of the fit). **d** Dissociation rates ($k_{off}$) of GST–dynein$_{331}$ in the absence and presence of increasing She1 concentrations. Off rates represent the inverse of the time constant from exponential fits to dwell-time distributions as reported in Fig. 6g (error bars, standard error)

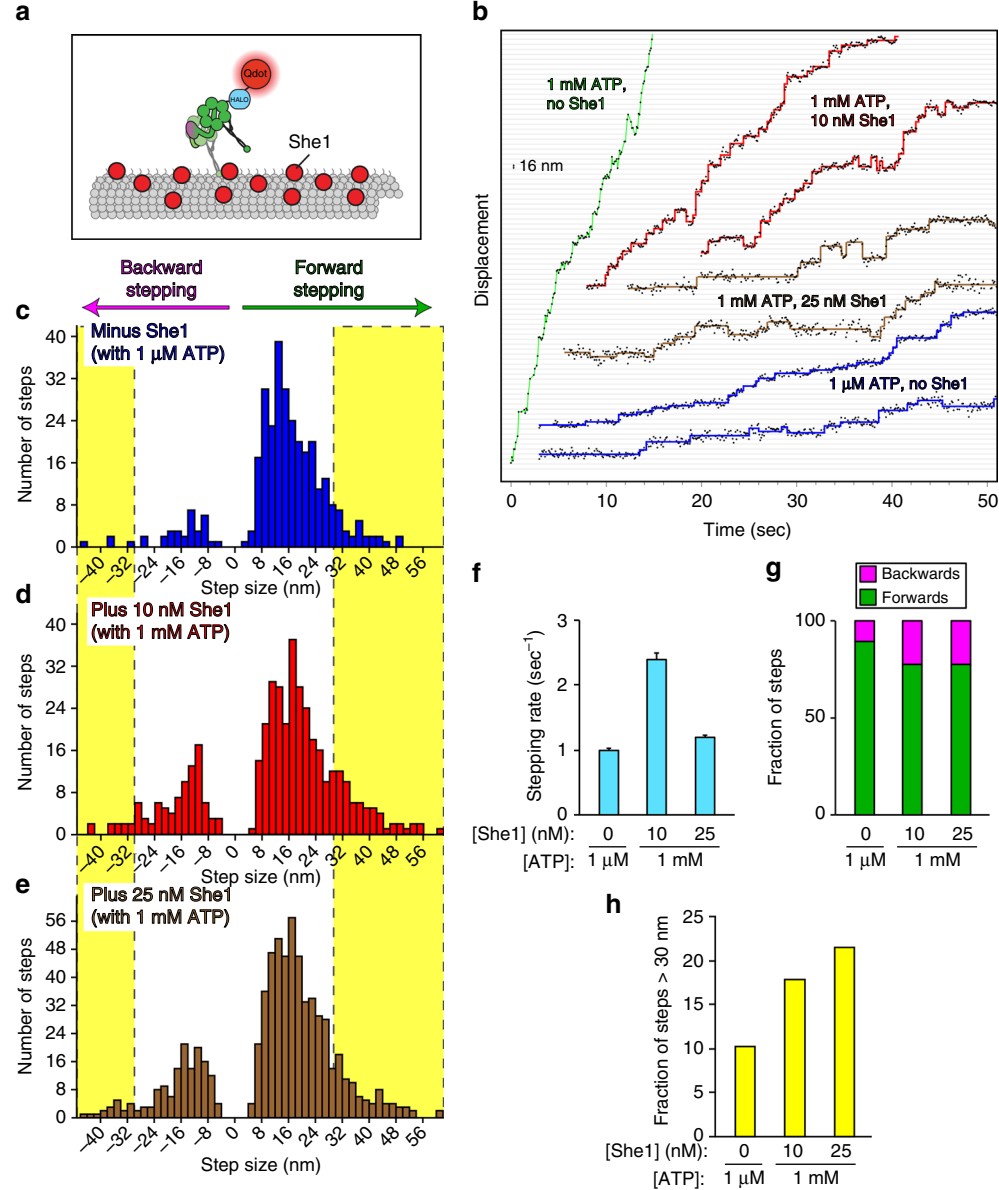

**Fig. 2** She1 reduces stepping frequency of dynein, and increases fraction of backward and large steps. **a** Schematic of experimental setup. **b** Representative traces of GST–dynein$_{331}$ movement tracked with high precision in the presence of 1 mM ATP (green), 1 μM ATP (blue), or 1 mM ATP and either 10 nM She1 (red) or 25 nM She1 (brown), as indicated. Steps were detected using custom-written code (see Methods). **c–e** Histograms of step size distributions for GST–dynein$_{331}$ in the absence or presence of She1, and with either 1 μM or 1 mM ATP, as indicated (yellow boxes delineate steps >30 nm in either direction; see **h**). **f** Histograms of dwell times between steps (Supplementary Fig. 1c) were fit to a convolution of two exponential functions with equal decay constants, which are plotted here as stepping rate (error bars, standard error of the fit). **g, h** The fraction of forward (i.e., minus end-directed) and backward (i.e., plus end-directed) steps (**g**), or large steps (**h**; in either the plus, or minus end direction; see yellow boxes in **c–e**) of GST–dynein$_{331}$ in the absence or presence of the indicated concentrations of She1 and ATP ($n = 320$ steps from 4 motors for no She1; 419 steps from 10 motors for 10 nM She1; 571 steps from 8 motors for 25 nM She1; Supplementary Fig. 1)

(i.e., E-hooks) from microtubules using the protease subtilisin (Fig. 3a,b). Although She1 was no longer able to bind to these microtubules (Fig. 3c), dynein was capable of binding and walking along them (Fig. 3d). Consistent with previous findings[30], addition of 10 nM She1 was sufficient to drastically alter dynein motility on undigested control microtubules (Fig. 3d–f, " + E-hooks"; Supplementary Fig. 2). However, in stark contrast to control microtubules, dynein motility on subtilisin-treated microtubules was completely unaffected by the presence of She1 (Fig. 3d–f, "− E-hooks"; Supplementary Fig. 2). Thus, microtubule binding by She1 is indeed required for it to affect dynein motility.

**She1 binds directly to the dynein motor domain**. Although the mechanism by which She1 affects dynein motility is unknown, previous single-molecule data suggested that She1 and dynein may interact along microtubules[30]. However, direct evidence for an interaction between these two molecules is lacking. To test whether the two molecules interact directly, we took advantage of the fact that dynein, but not She1, is able to bind to subtilisin-treated microtubules (Fig. 3). If the two molecules interact, then microtubule-bound dynein would recruit She1 to the microtubule, and this binding could be observed and quantitated by total internal reflection fluorescence (TIRF) microscopy (Fig. 4a). A fixed concentration of fluorescent She1-TMR (40 nM) was incubated with

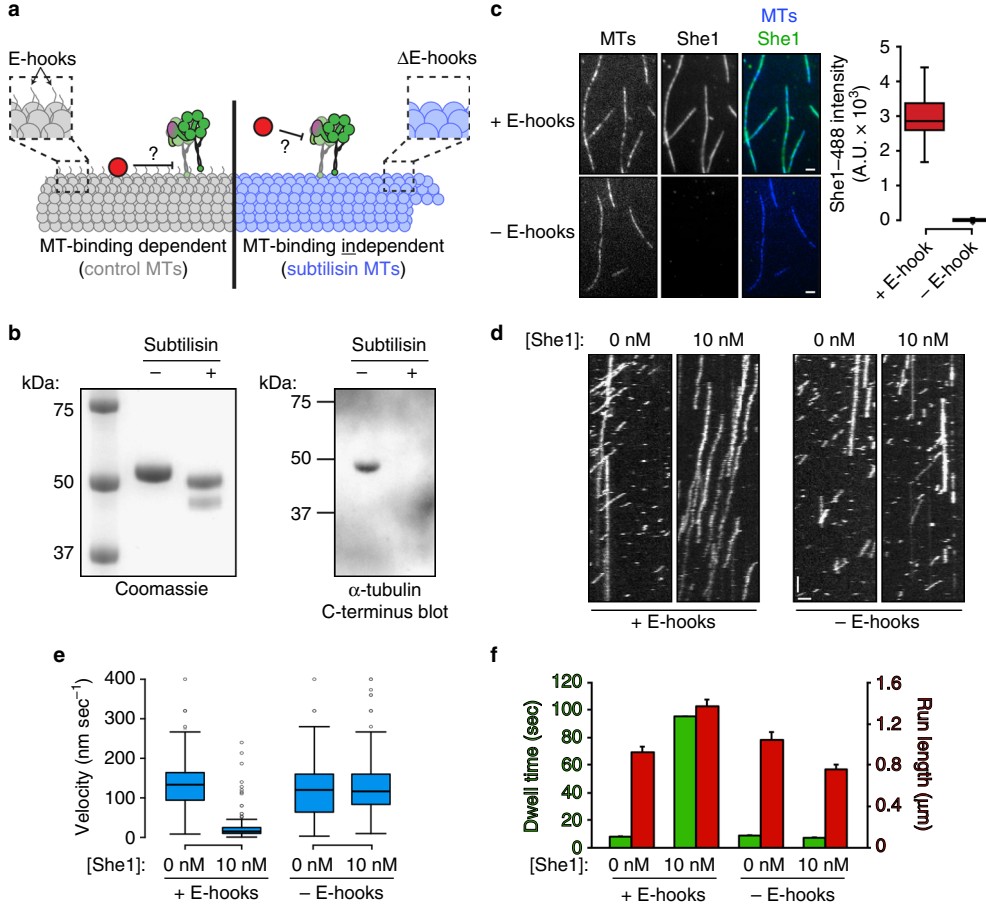

**Fig. 3** Microtubule binding by She1 is required for it to affect dynein motility. **a** Schematic depicting two distinct mechanisms by which She1 may affect dynein motility. She1 either requires its microtubule-binding activity to affect dynein motility (left), or it affects dynein independently of its microtubule-binding activity (right). **b** Coomassie-stained SDS–polyacrylamide gel (left) and immunoblot (anti-alpha-tubulin-C-terminus; right) of taxol-stabilized HiLyte647-labeled microtubules incubated with or without subtilisin, as indicated (see Methods). **c** 10 nM She1-TMR was incubated with either control ("+E-hooks") or subtilisin-digested ("−E-hooks") coverslip-immobilized HiLyte647-labeled microtubules for 5 min, then images were acquired by TIRF microscopy. Representative fluorescence images are shown (left) along with box plots of microtubule-bound She1-TMR fluorescence intensity values (scale bars, 2 μm). **d** Representative kymographs showing GST–dynein$_{331}$ motility in the absence or presence of 10 nM She1 on either control or subtilisin-digested microtubules. Note that for each experiment in which She1 is included, She1 was pre-incubated with the microtubules for 5 min before addition of GST–dynein$_{331}$, which was diluted in motility mix that also included 10 nM She1 (horizontal scale bar, 2 μm; vertical scale bar, 1 min). **e** Box plot of GST–dynein$_{331}$ velocity values in indicated conditions. **f** Mean run lengths (red) and dwell times (green) for GST–dynein$_{331}$ molecules along either control or subtilisin-digested microtubules in the absence or presence of She1 (error bars, standard error of the mean; n ≥ 199 individual motors for each condition). For box-whisker plots in **c** and **e**, whiskers define the range, boxes encompass 25th to 75th quartiles, lines depict the medians, and circles depict outlier values (defined as values greater than (upper quartile + 1.5 × interquartile distance), or less than (lower quartile − 1.5 × interquartile distance); Supplementary Fig. 2)

subtilisin-digested microtubules in the absence or presence of increasing concentrations of a monomeric, non-processive, GFP-tagged dynein motor domain fragment (GFP–dynein$_{331}$; Supplementary Fig. 3b). We observed robust recruitment of She1 to subtilisin-digested microtubules by increasing concentrations of the dynein motor domain, thus demonstrating a direct interaction between She1 and dynein (Fig. 4b,c).

The aforementioned binding experiment (Fig. 4c) was performed in the absence of nucleotide. In these conditions, dynein adopts a conformation in which the linker is in the post-powerstroke state and the MTBD is in a high microtubule-binding affinity state[34–36, 39, 42]. To determine if She1 preferentially binds to a particular dynein conformational state, we repeated the binding experiment with either no nucleotide (as above) or with ATP and vanadate ($V_i$). The latter traps dynein in an ADP–$V_i$ intermediate (ADP-$P_i$ mimic) in which the linker is in the pre-powerstroke state and the MTBD is in the low

microtubule-binding affinity state[35, 37, 39] (Fig. 4d). To correct for the differential microtubule-binding affinity of dynein in the absence of nucleotide vs. in the presence of ATP + $V_i$ (Supplementary Fig. 3a), we correlated the degree of microtubule binding by GFP–dynein$_{331}$ in each condition to the extent of She1 microtubule recruitment (i.e., fluorescence intensity of GFP–dynein$_{331}$ vs. She1-TMR). We found that for a given degree of GFP–dynein$_{331}$ microtubule binding, more She1 was recruited to microtubules in the absence of nucleotide than in the presence of ATP + $V_i$ (Fig. 4e). These data suggest that She1 has a higher affinity for dynein in the apo state than in the ADP–$V_i$ state. Moreover, they indicate that She1 recognizes a structural feature of dynein that undergoes a nucleotide-induced conformational change.

Although our findings indicate that She1 binds preferentially to one conformation over another, She1 was indeed able to bind to dynein in both nucleotide states. If true we reasoned that She1

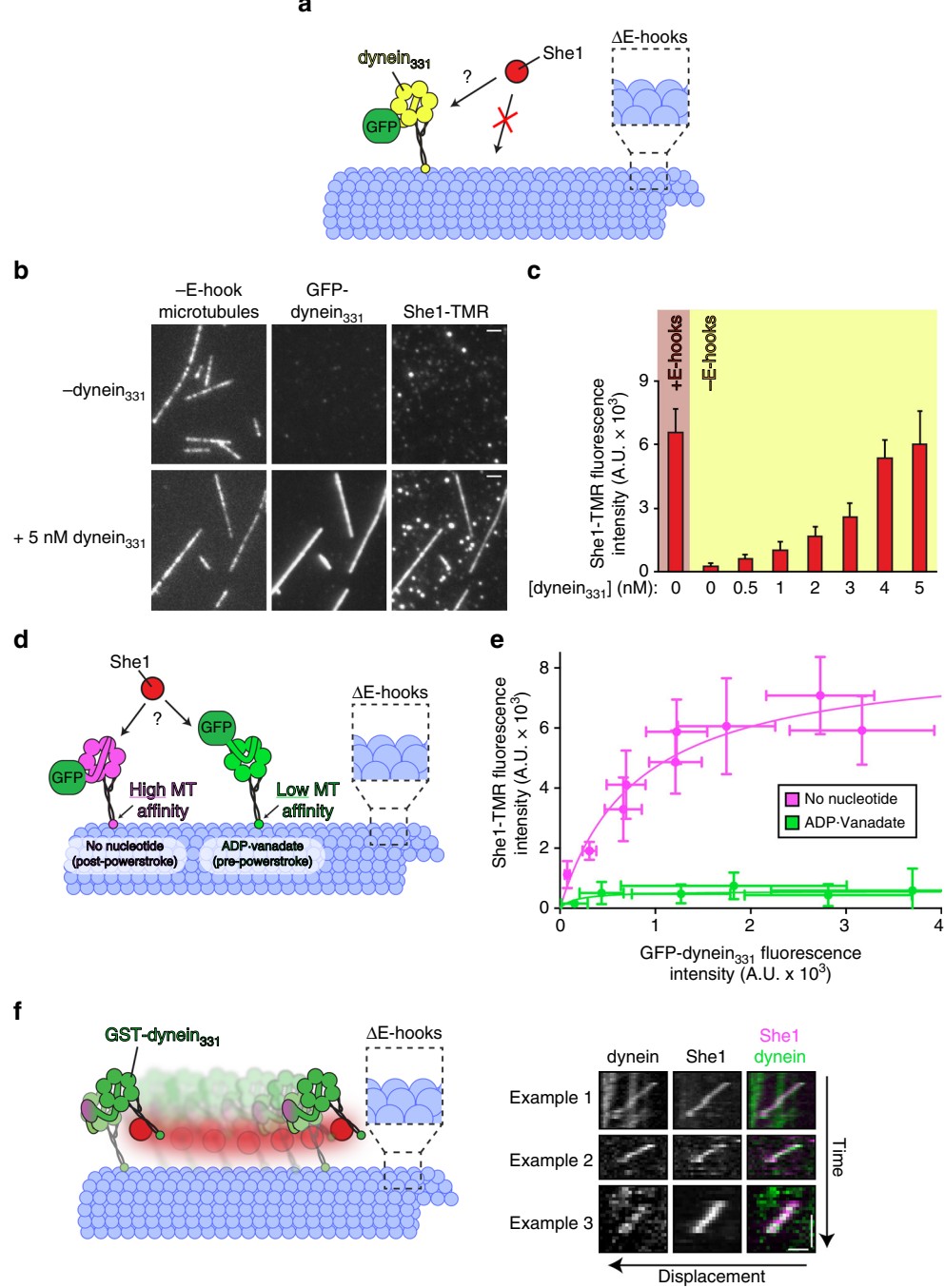

**Fig. 4** She1 binds directly to dynein, and recognizes a specific nucleotide-bound state. **a** Schematic of experimental setup. Note that for these experiments, a GFP-tagged monomeric dynein$_{331}$ (GFP–dynein$_{331}$) fragment was used. **b–d** Representative fluorescence images (**b**) and quantitation (**c**) of She1-TMR recruitment (fixed at 40 nM) to control ("+E-Hook") or subtilisin-digested ("−E-hook") microtubules by increasing concentrations of GFP–dynein$_{331}$ (scale bars, 2 µm; error bars, standard deviation; $n \geq 19$ microtubules, and $\geq 75$ µm of MT length for each condition). **d** Schematic of experimental setup. Note that the absence of nucleotide elicits a conformational state that is distinct from that of dynein in the presence of ATP and vanadate (see text). **e** Relative recruitment of She1-TMR by GFP–dynein$_{331}$ in the presence of either no nucleotide (apo) or 3 mM ATP and vanadate (ADP–vanadate). Different points reflect the mean fluorescence intensity values (along with standard deviations) of She1-TMR (fixed at 40 nM) vs. increasing concentrations of GFP–dynein$_{331}$. Given the different microtubule-binding affinity of GFP–dynein$_{331}$ in each nucleotide state (Supplementary Fig. 3a), the extent of She1-TMR microtubule recruitment was directly compared to the relative microtubule binding by GFP–dynein$_{331}$ ($n \geq 10$ microtubules, and $\geq 36$ µm of MT length for each condition). **f** Cartoon (left) and three example kymographs (right) depicting that on subtilisin-digested microtubules, She1 remains bound to GST–dynein$_{331}$ as it walks, and thus transitions through many iterations of its mechanochemical cycle (horizontal scale bar, 1 µm; vertical scale bar, 30 s; Supplementary Fig. 3)

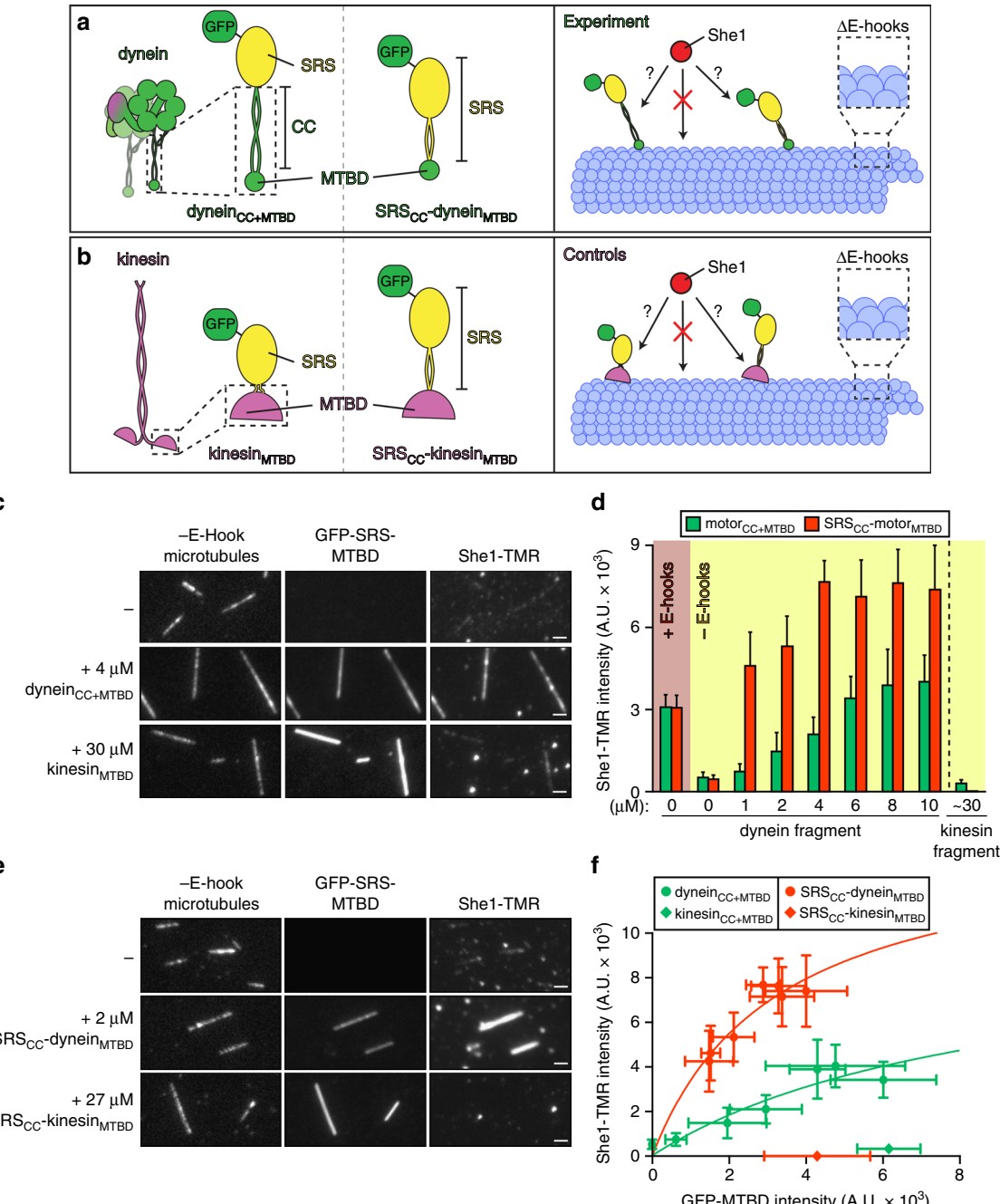

**Fig. 5** She1 binds directly to the dynein microtubule-binding domain. **a–b** Cartoon representations of the various GFP–seryl tRNA synthetase (SRS)–dynein (**a**) and kinesin (**b**) fusions used in the microtubule recruitment assays (left) along with a schematic of the experimental setup (right). The SRS globular domain fused to either the dynein coiled coil (CC) and microtubule-binding domain (MTBD), or the kinesin MTBD, respectively are depicted in **a** and **b**, left, while **a** and **b**, middle, depict the SRS globular and coiled coil domains fused to either the dynein or kinesin MTBD domains, respectively. **c, e** Representative fluorescence images of She1-TMR recruitment to subtilisin-digested microtubules by GFP–SRS–dynein$_{CC+MTBD}$ (**c**) or GFP–SRS–SRS$_{CC}$–dynein$_{MTBD}$ (**e**), but not the respective kinesin MTBD controls. Respective images acquired from each experiment are displayed with identical brightness and contrast levels. Note that in spite of the lesser degree of microtubule-binding by the SRS$_{CC}$–dynein$_{MTBD}$ fusion (in **e**) compared to dynein$_{CC+MTBD}$ (in **c**), more She1-TMR is recruited to microtubules by the former (scale bars, 2 μm). **d** Quantitation of the extent of She1-TMR recruitment to subtilisin-digested microtubules by increasing concentrations of the indicated GFP–SRS–MTBD fusion (error bars, standard deviation; $n \geq 19$ microtubules, and ≥82 μm of MT length for each condition). **f** Relative recruitment of She1-TMR by indicated GFP–SRS–MTBD fusion. Different points reflect the mean fluorescence intensity values (along with standard deviations) for She1-TMR (fixed at 20 nM) vs. increasing concentrations of indicated GFP–SRS–MTBD fusions. Note that concentrations of the kinesin$_{MTBD}$ fusions were chosen such that the degree of their microtubule binding closely matched the maximal microtubule binding by the corresponding dynein fragment. As in Fig. 4e, the extent of She1-TMR microtubule recruitment was directly compared to relative microtubule binding by each GFP–SRS–MTBD fragment; Supplementary Fig. 3)

would stay bound to dynein as it walked along subtilisin-treated microtubules (i.e., those to which She1 is unable to bind) and thus progressed through many iterations of its mechanochemical cycle. Consistent with this notion, we observed several examples of such events in which She1-TMR was observed colocalizing with moving single molecules of GST–dynein$_{331}$ (Fig. 4f; only 14 such events were observed out of several hundred moving dynein molecules). Thus, in spite of its preferred affinity for the apo state, She1 can indeed remain bound to dynein throughout its entire mechanochemical cycle.

**She1 binds directly to the dynein microtubule-binding domain.** Since microtubule binding by She1 is required for it to affect dynein motility (Fig. 3), and She1 and dynein interact directly (Fig. 4), we reasoned that She1 might exert its effect on dynein motility by binding to a surface of the motor domain that is in close proximity to the microtubule. To test this hypothesis, we generated recombinant protein fragments that encompass the dynein MTBD and the coiled coil (CC; which links the AAA ring to the MTBD) fused to seryl tRNA synthetase (SRS; Fig. 5a, left). It has been shown that a nearly identical fusion protein derived

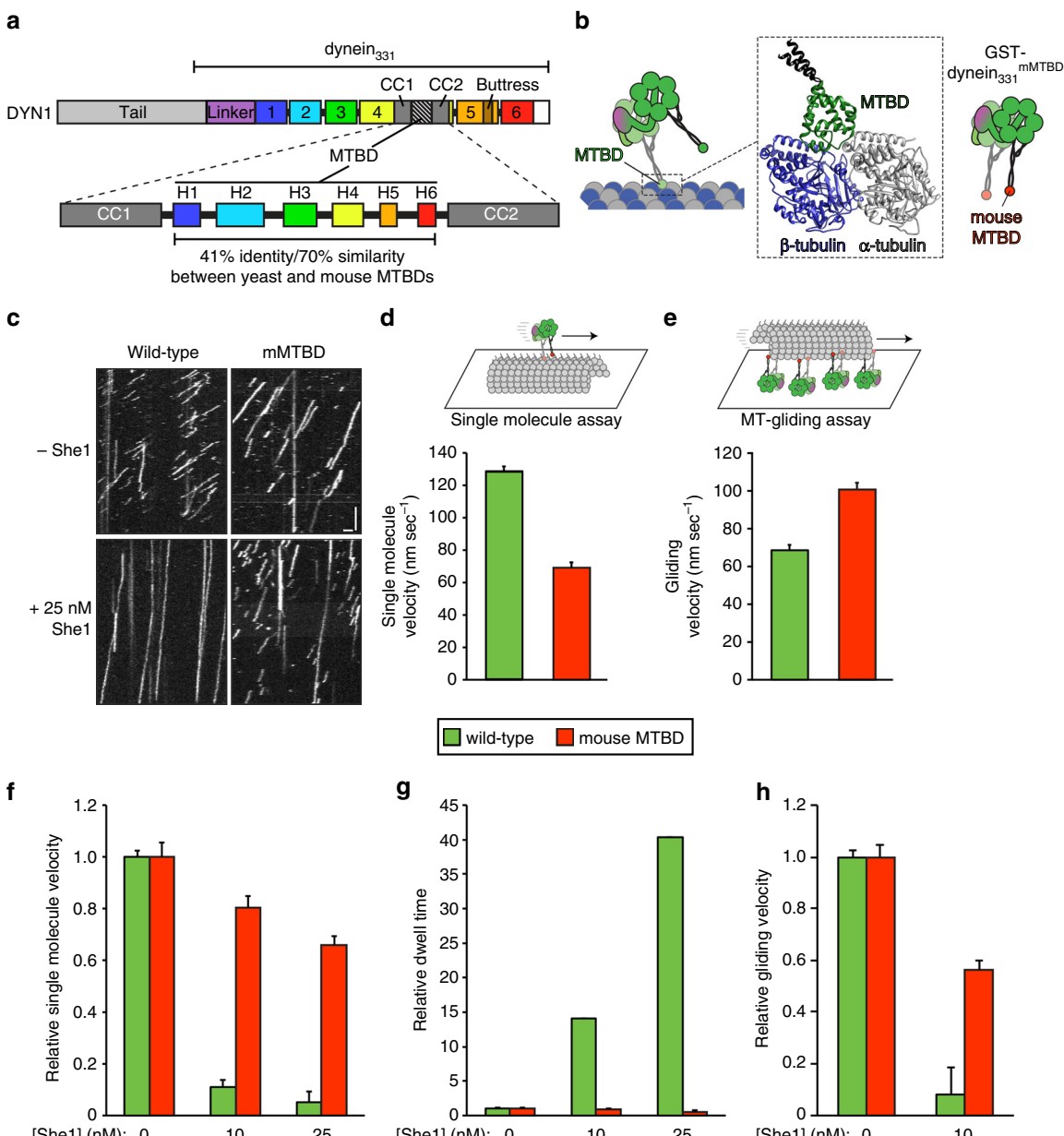

**Fig. 6** A dynein motor with a mutated MTBD exhibits reduced She1 sensitivity. **a** Schematic representation of the yeast dynein heavy chain (*DYN1*) with domain structure indicated (domains 1–6 represent the individual AAA domains; CC coiled coil; H1–H6, helices that comprise the MTBD). **b** Cartoon representation with homology model of the yeast dynein MTBD bound to alpha and beta-tubulin (green, MTBD; dark gray, CC1 and CC2; generated using one-to-one threading of yeast *DYN1* sequence into 3J1T[45] on the Phyre2 server[71]). **c** Kymographs depicting single-molecule motility of GST–dynein$_{331}$ and GST–dynein$_{331}$$^{mMTBD}$ in the absence (top) or presence (bottom) of 25 nM She1 (horizontal scale bar, 2 μm; vertical scale bar, 1 min). **d**, **e** Plots depicting mean velocity of GST–dynein$_{331}$ and GST–dynein$_{331}$$^{mMTBD}$ in single-molecule (**d**) and ensemble microtubule gliding assays (**e**; error bars, standard error). **f**, **g** Plots depicting effects of She1 on the relative velocity (**f**) and dwell time (**g**) of single molecules of GST–dynein$_{331}$ and GST–dynein$_{331}$$^{mMTBD}$. **h** Plot depicting effects of She1 on the relative microtubule gliding velocity of coverslip-immobilized GST–dynein$_{331}$ and GST–dynein$_{331}$$^{mMTBD}$ (error bars, standard error; $n \geq 147$ individual motors for each condition for single-molecule assay; $n \geq 21$ microtubules for each condition for the ensemble motility assay); Supplementary Figs. 4 and 5)

from mouse dynein adopts a native fold and retains microtubule-binding activity[40], [45], [47]. We expressed and purified this dynein fragment (dynein$_{CC+MTBD}$) from bacteria (Supplementary Fig. 3b) and performed the microtubule recruitment assay described above with subtilisin-treated microtubules. As a control we generated an SRS fusion that is linked to the MTBD of human kinesin-1 via a flexible linker (kinesin$_{MTBD}$; Fig. 5b, left, and Supplementary Fig. 3b). Consistent with the notion that She1 binds to a region of dynein that is in close proximity to the

microtubule, we found that dynein$_{CC+MTBD}$, but not kinesin$_{MTBD}$, recruited She1 to microtubules in a concentration dependent manner, thus demonstrating a direct interaction between She1 and dynein$_{CC+MTBD}$ (Fig. 5c, d, green bars).

To further refine the She1 binding surface within dynein, we generated an SRS fusion construct that included only the 124 amino acid dynein MTBD. To best ensure the MTBD adopted a native fold and retained microtubule-binding activity, we replaced the native dynein CC with one from SRS

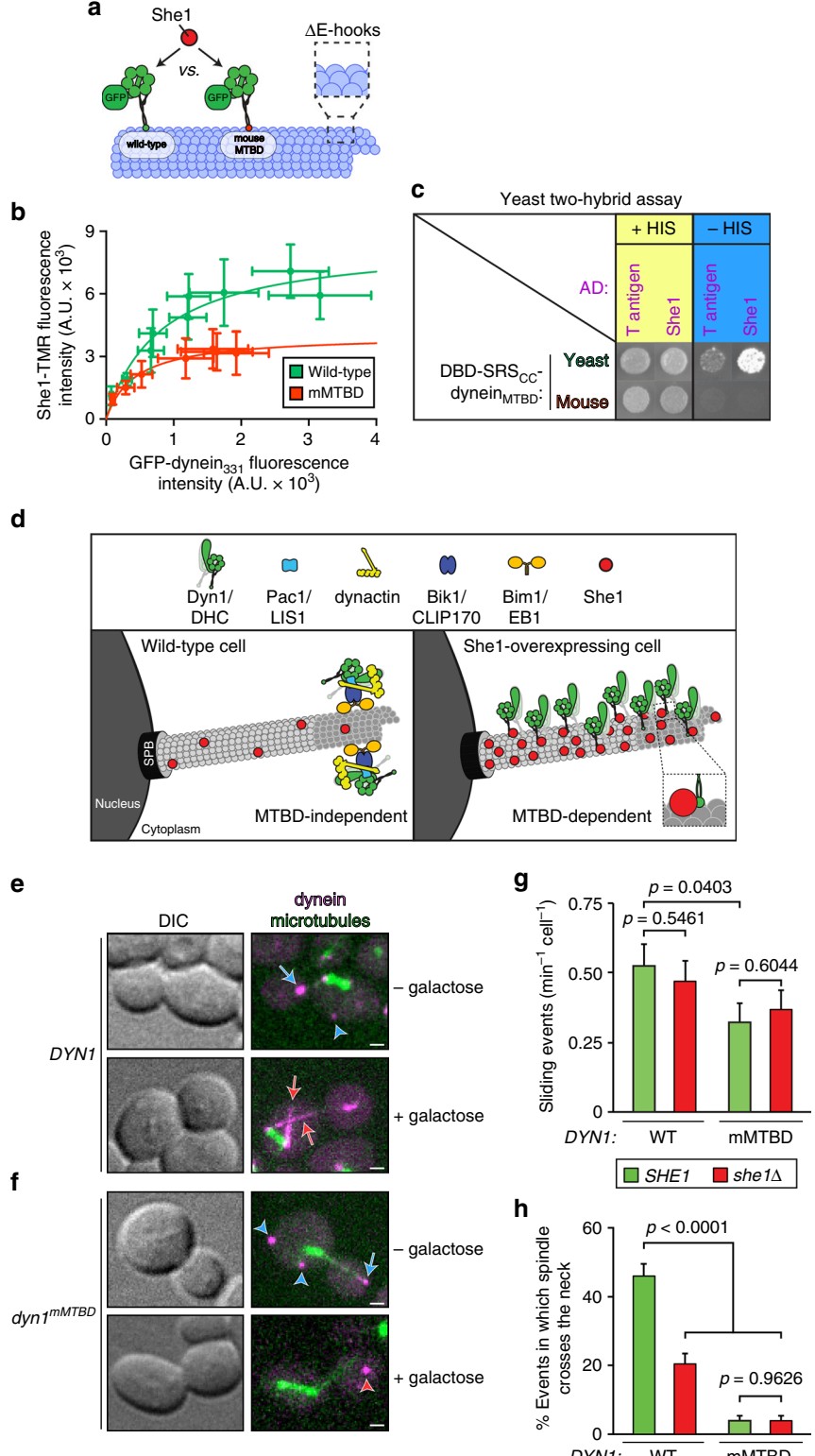

(SRS$_{CC}$–dynein$_{MTBD}$; Fig. 5a, middle and Supplementary Fig. 3b, c). To rule out the possibility that She1 microtubule recruitment was being mediated by the SRS CC, we generated a similar fusion protein that included the kinesin MTBD in place of the dynein MTBD (SRS$_{CC}$–kinesin$_{MTBD}$; Fig. 5b, middle, and Supplementary Fig. 3b). We found that SRS$_{CC}$–dynein$_{MTBD}$, but not SRS$_{CC}$–kinesin$_{MTBD}$, was sufficient to robustly recruit She1 to microtubules, indicating that She1 directly contacts the dynein MTBD (Fig. 5d, red bars and Fig. 5e).

As above, we correlated the degree of microtubule binding by SRS$_{CC}$–dynein$_{MTBD}$ and dynein$_{CC+MTBD}$ to the extent of She1 microtubule recruitment by each (i.e., fluorescence intensity of GFP–SRS fusion vs. She1-TMR). This revealed that for a given degree of microtubule binding, more She1 was recruited to microtubules by SRS$_{CC}$–dynein$_{MTBD}$ than by dynein$_{CC+MTBD}$, thus indicating that She1 has a higher affinity for the former, in spite of the latter encompassing a larger region of dynein (Fig. 5f). Given the difference in apparent affinity of She1 for dynein in the apo vs. ADP–V$_i$ state (Fig. 4e), we hypothesized that the difference in She1 binding affinity for the two different SRS fusion proteins was due to possible differences in the conformation of the MTBD. It is well established that the MTBD undergoes conformational changes in response to its nucleotide and microtubule-bound state[37, 42, 43, 45, 47]. The structural plasticity of this domain allows the motor to cycle through periods of high (in its apo and ADP-bound state) and low (in its ATP and ADP-P$_i$) microtubule-binding affinity during processive runs. We found that SRS$_{CC}$–dynein$_{MTBD}$ exhibited a ~9-fold higher microtubule-binding affinity than dynein$_{CC+MTBD}$ (Supplementary Fig. 3d; $0.9 \pm 0.1\,\mu M$ vs. $7.8 \pm 3.0\,\mu M$; $\pm$ SE of fit) which indicates that the two MTBD fusions are indeed in distinct conformational states. These data also confirm that She1 exhibits higher affinity for the dynein MTBD in its high microtubule-binding affinity conformation (see Supplementary Fig. 4a and Discussion).

To confirm the interaction between She1 and the dynein MTBD, we performed a yeast two-hybrid assay. We expressed various DNA-binding domain (GAL4–DBD) fusions (i.e., GAL4–DBD–dynein$_{CC+MTBD}$, GAL4–DBD–SRS$_{CC}$–dynein$_{MTBD}$, or GAL4–DBD–SRS$_{CC}$–kinesin$_{MTBD}$) along with either a transcriptional activation domain (GAL4–AD)–She1, or negative control (GAL4–AD–large T antigen) fusion in yeast cells harboring GAL4 responsive reporter genes. Positive interactions are detected by growth on histidine-deficient media. Consistent with our in vitro data, this analysis revealed an interaction between She1 and SRS$_{CC}$–dynein$_{MTBD}$; however, we observed no detectable two-hybrid interaction between She1 and either SRS$_{CC}$–kinesin$_{MTBD}$ or dynein$_{CC+MTBD}$ (Supplementary Fig. 3e),

the latter of which is consistent with a significantly weaker interaction as determined by our in vitro assay (Fig. 5f).

We next asked whether She1 exhibits any affinity for regions of dynein outside the MTBD. To this end, we performed a recruitment assay on undigested ("+E-hook") microtubules. Incubation of microtubules with high concentrations of She1 (40 nM) and a dynein mutant (45 nM) lacking its MTBD (GFP–dynein$_{331}^{\Delta MTBD}$, Supplementary Fig. 3b) resulted in no apparent microtubule recruitment of GFP–dynein$_{331}^{\Delta MTBD}$ by She1 (Supplementary Fig. 3f, g). Taken together our results indicate that She1 binds exclusively to the dynein MTBD.

**A dynein motor with reduced sensitivity to She1.** If She1 indeed affects dynein motility through interactions with the dynein MTBD, then we reasoned that mutations within this region that reduce She1 binding would also disrupt any She1-mediated effects on dynein motility. Thus, we sought to introduce mutations within the MTBD that would disrupt She1 binding. Rather than generate a library of random mutants that would potentially disrupt MTBD structure or function (e.g., microtubule-binding activity), we instead developed a strategy in which the dynein MTBD from an evolutionarily distant organism was used to replace that from yeast *DYN1* (dynein heavy chain), thus generating a chimeric dynein MTBD mutant. We hypothesized that She1 may exhibit binding specificity for yeast dynein and may therefore exhibit reduced binding to metazoan dynein. To test this possibility, we generated a chimeric GST–dynein$_{331}$ fragment in which only the globular MTBD was replaced by the corresponding MTBD from mouse dynein (GST–dynein$_{331}^{mMTBD}$; Fig. 6a, b). Sequence analysis revealed 41% identity and 70% similarity between yeast and mouse dynein MTBDs, indicating significant divergence in primary sequence between the two motors (Fig. 6a; Supplementary Fig. 4b). To our surprise, the GST–dynein$_{331}^{mMTBD}$ chimera was capable of walking along microtubules, albeit with slightly altered motility parameters with respect to wild-type GST–dynein$_{331}$ (Fig. 6c–e; and Supplementary Fig. 5). Specifically, GST–dynein$_{331}^{mMTBD}$ walked at roughly half the velocity in single-molecule assays (68.9 nm/s vs. 128 nm/s), but moved microtubules faster than wild-type dynein in an ensemble microtubule gliding assay (using equivalent concentrations of coverslip-immobilized motors; Methods). Moreover, the chimeric mutant walked longer distances and spent more time bound to microtubules than the wild-type motor in single-molecule assays (Supplementary Fig. 5).

Consistent with the notion that She1 makes contacts with the MTBD, GST–dynein$_{331}^{mMTBD}$ exhibited reduced sensitivity to She1 in terms of its effects on velocity (in single-molecule and

**Fig. 7** Dynein$^{mMTBD}$ exhibits reduced affinity for She1 in vitro and is She1-insensitive in vivo. **a** Schematic representation of the experimental setup used in **b**. **b** Relative recruitment of She1-TMR by monomeric GFP-dynein$_{331}$ or GFP–dynein$_{331}^{mMTBD}$. Different points reflect the mean fluorescence intensity values (along with standard deviations) for She1-TMR (fixed at 40 nM) vs. increasing concentrations (0–30 nM for wild-type and 0–100 nM for mMTBD) of indicated GFP-dynein$_{331}$. **c** Two-hybrid assay demonstrating an interaction between the yeast derived dynein$_{MTBD}$ and She1 (Methods). **d** Cartoon representation of the localization of full-length dynein heavy chain (Dyn1) in either wild-type (left) or She1-overexpressing cells (right). Note that the mechanism for plus end localization of dynein (which is MTBD-independent[51]) is distinct from that by which dynein binds along the length of astral microtubules upon She1 overexpression (MTBD-dependent; Supplementary Fig. 6). **e, f** Representative images of *GAL1p:SHE1* cells expressing mRuby2-Tub1 (α-tubulin) and either Dyn1-3YFP (**e**) or Dyn1$^{mMTBD}$-3YFP (**f**). Cells were grown to mid-log phase in SD media supplemented with raffinose (uninduced; −galactose) or galactose plus raffinose (induced for 3.5 h; +galactose) and then mounted on agarose pads for confocal fluorescence microscopy (blue arrows, plus end foci; blue arrowheads, cortical foci; red arrows, astral microtubule decoration; red arrowhead, cytoplasmic focus). Foci were identified in two-color movies and scored accordingly (scale bars, 1 μm). **g, h** Dynein-mediated spindle movements were quantitated in hydroxyurea (HU)-arrested *kar9Δ* cells with indicated *DYN1* and *SHE1* alleles. Cells were arrested with HU for 2.5 h, and then mounted on agarose pads containing HU for confocal fluorescence microscopy. Full Z-stacks of GFP-labeled microtubules (GFP-Tub1) were acquired every 10 s for 10 min. Cells with buds of at least 2.5 μm in diameter were chosen for analysis. Graphs depicting the number of dynein-mediated spindle movements (**g**) and the fraction of such events in which the spindle traversed the bud neck (**h**; in which the spindle midpoint crossed the bud neck) for the indicated yeast strains are shown (error bars, standard error of proportion; $n \geq 43$ cells; $n \geq 155$ events). *P*-values were calculated using a two-tailed unpaired *t* test

ensemble assays; Fig. 6f,h) and dwell time (Fig. 6g). We used our microtubule recruitment assay to compare the relative affinity of She1 for monomeric GFP–dynein$_{331}$ and GFP–dynein$_{331}$$^{mMTBD}$ (Fig. 7a; Supplementary Fig. 3b), and found that the reduced effects of She1 on GST–dynein$_{331}$$^{mMTBD}$ motility were indeed due to compromised She1–dynein binding. Although GFP–dynein$_{331}$$^{mMTBD}$ was capable of recruiting She1 to subtilisin-treated microtubules, the relative degree of recruitment was lower than that of the wild-type motor domain (Fig. 7b), indicating a significantly lowered affinity of She1 for the chimeric motor. We confirmed the reduced affinity of She1 for the mouse dynein MTBD using the two-hybrid assay, which revealed no detectable two-hybrid interaction between She1 and a mouse dynein variant of the SRS$_{CC}$–dynein$_{MTBD}$ fragment (Fig. 7c).

**Dynein$^{mMTBD}$ mutant cells exhibit She1-insensitive phenotypes.** Overexpression of She1 in yeast leads to defects in dynein pathway function as is apparent by errors in spindle positioning (Supplementary Fig. 7a) and synthetic genetic interactions with *KAR9*[30], the latter of which functions in a parallel spindle orientation pathway[48]. Although the precise cause for dynein dysfunction in these cells is unclear, She1 overexpression leads to a relocalization of dynein from microtubule plus ends (Fig. 7d, left, and e, blue arrow)—from where it is offloaded to Num1 cortical receptor sites (Fig. 7e, blue arrowhead)—to along the length of astral microtubules (Fig. 7d, right and Fig. 7e, red arrows). We hypothesized that this relocalization may be a consequence of She1 enhancing dynein's microtubule-binding affinity via direct interactions between astral microtubules (Fig. 3) and the dynein MTBD (Fig. 5). To distinguish between this possibility and one in which the relocalization is a consequence of a redistribution of the dynein plus end targeting complex (which is comprised of Bik1, Pac1, and Bim1 in yeast[31, 49, 50]), we assessed: (1) whether the dynein MTBD, which is dispensable for plus end targeting[51], is required for the relocalization phenotype, and (2) whether Pac1, which is necessary for plus end targeting[31], is required for this phenotype (Supplementary Fig. 6a). For these experiments, we assessed dynein localization (either wild-type Dyn1-3YFP, or Dyn1$^{ΔMTBD}$-3YFP) in *GAL1p:SHE1* cells grown in either the absence or presence of galactose, a potent stimulant of the *GAL1* promoter (*GAL1p*). Consistent with the notion that the relocalization phenotype is a consequence of She1 enhancing dynein's microtubule-binding affinity, we found that deletion of the MTBD prevented dynein relocalization, whereas loss of Pac1 had no impact on the relocalization phenotype (Supplementary Fig. 6b, c).

Next, we asked whether the mouse MTBD chimera exhibits reduced sensitivity to She1 in cells. In the absence of She1 overexpression, dynein$^{mMTBD}$ localizes to microtubule plus ends and the cell cortex in a manner similar to that of wild-type dynein (Fig. 7f, blue arrows and arrowheads). Consistent with the notion that dynein$^{mMTBD}$ is less sensitive to She1, it was not redistributed along astral microtubules upon She1 overexpression (Fig. 7f). In spite of this, we noted that its plus end and cortical localization were reduced with respect to cells not overexpressing She1, and there appeared to be cytoplasmic aggregates of dynein$^{mMTBD}$ (Fig. 7f, red arrowhead). Although the basis for this mislocalization is unclear, it is the likely basis for the prevalence of misoriented spindles in these cells (Supplementary Fig. 7a).

Although loss of She1 does not lead to a significant spindle mispositioning defect (Supplementary Fig. 7b), She1 has been implicated in polarizing dynein-mediated spindle movements toward the daughter cell. Specifically, cells deleted for She1 exhibit a reduced fraction of dynein-mediated spindle movements

that result in the spindle traversing the mother-bud neck in a spindle oscillation assay[30]. In this assay, the movements of pre-anaphase spindles are monitored in *kar9Δ* hydroxyurea (HU)-arrested cells, the latter of which eliminates spindle movements due to spindle elongation during anaphase. Deletion of *KAR9* leads to an enhancement of dynein-mediated spindle movements[52, 53] and also eliminates any movements that might be mediated by the *KAR9* pathway for spindle orientation.

Although dynein$^{mMTBD}$ appeared to possess nearly wild-type activity as assessed by a single time point spindle positioning assay (Supplementary Fig. 7b; Methods), and was capable of mediating spindle movements in the spindle oscillation assay, the frequency of these movements was reduced to ~61% of wild-type (Fig. 7g). Moreover, we noted that the fraction of dynein-mediated spindle movements that resulted in neck crossing was greatly reduced in the *dyn1$^{mMTBD}$* cells (Fig. 7h). Although deletion of She1 reduced neck crossing by 53% in *DYN1* (wild-type dynein) cells, deletion of She1 had no additional impact on the degree of neck crossing in *dyn1$^{mMTBD}$* cells (Fig. 7h; Supplementary Fig. 7c). Taken together, our data indicate that the mouse MTBD chimera indeed exhibits reduced sensitivity to She1, and further confirm that the MTBD is the main site of interaction for She1.

## Discussion
Our study provides the first detailed molecular dissection of the mechanism by which a MAP can affect the function of a microtubule motor. Specifically, we have found that She1 affects dynein motility by increasing its microtubule-binding affinity (as a consequence of reducing its microtubule dissociation rate; Fig. 1d), which causes a reduction in stepping frequency (Fig. 2b, f) and consequent ATP turnover (Fig. 1b, c). These effects are due to the simultaneous and direct interactions between She1, the microtubule (via the C-terminal tails of tubulin), and the small (124 amino acids) globular dynein MTBD (Fig. 5). In light of the fact that She1 and dynein directly interact, we can extract an approximate She1–dynein binding affinity from the She1 concentration value at which dynein velocity is half-maximally reduced: 0.17 nM[30]. To our knowledge this is the first time that any such regulatory molecule has been shown to contact the dynein MTBD. Although She1 is a yeast-specific dynein regulatory factor, it may define a new class of motor regulatory MAP. Moreover, our work identifies the dynein MTBD as a target for MAP-mediated dynein regulation.

She1 is the first molecule identified to date that has the capacity to alter dynein stepping behavior (i.e., increases the frequency of large and backward steps; Fig. 2g, h). Although the reasons for this are unclear, we hypothesize that these changes in stepping behavior are a consequence of one of the motor heads within a dimer becoming unbound from She1 for brief periods of time. In such a scenario, one motor head unbinds from microtubule-bound She1 and steps forward. Given the lower likelihood of the lagging She1-bound head unbinding from the microtubule (due to reduced dissociation rates; Fig. 1d), the leading She1-unbound head in this scenario will unbind from the microtubule and consequently steps backward. Alternatively, given that the leading head is generally less likely to detach from the microtubule at increased interhead separations (due to tension exerted on the linker[54]), the lagging She1-bound head may eventually detach at sufficiently large interhead separations, which may result in larger than normal step sizes. Simultaneous two-color imaging of both heads will be required to understand the basis for the altered stepping behavior.

We found that She1 exhibits an enhanced affinity for dynein in the apo (nucleotide-free) state, during which the MTBD is in a

high microtubule-binding affinity state[42]. We observed this preferential binding in the context of the full motor domain (apo vs. ADP–vanadate; Fig. 4e) and with an isolated dynein MTBD fragment (SRS$_{CC}$–dynein$_{MTBD}$ vs. dynein$_{CC+MTBD}$; Fig. 5f and Supplementary Fig. 3d), the latter of which we confirmed using a yeast two-hybrid assay (Supplementary Fig. 3e). A previous study demonstrated a similar nucleotide-specific interaction between metazoan LIS1 and dynein. In this example, LIS1 was only found to interact with dynein in its ADP–V$_i$ state[55] (the same is not true for yeast dynein and the LIS1 homolog, Pac1, which interact in a nucleotide-independent manner[33]). Given the fact that Pac1 interacts with the dynein AAA ring (between AAA3 and AAA4[33]) the mechanism by which She1 recognizes the nucleotide state of dynein is therefore distinct. Structural studies have revealed the basis for differential microtubule-binding affinity of dynein in its various nucleotide-bound conditions. The largest conformational changes that take place in the MTBD when the motor undergoes changes in microtubule-binding affinity are the movement of helix 1 (H1, root mean square deviation of 10.1 Å; Fig. 6a and Supplementary Fig. 4a) and CC1 (RMSD = 8.1 Å)[45]. Thus, it is reasonable to hypothesize that She1 makes contacts with a region of the MTBD that encompasses these elements.

It is currently unclear what the relevance of this conformational specificity of She1 for dynein is, especially in light of the fact that She1 can remain bound to a walking dynein motor (along subtilisin-treated microtubules; Fig. 4f), which is undergoing many iterations of the mechanochemical cycle. One possibility may be that in the context of non-subtilisin-treated microtubules, She1 holds dynein to microtubules by locking the motor in its high microtubule-binding affinity state. In this model, upon encountering each other along microtubules, She1 would bind dynein in its apo (or ADP-bound) state, which is the predominant microtubule-bound state of dynein[43]. Given the high affinity interaction between dynein and She1 (<0.2 nM; see above), it is possible that even upon ATP binding, the dynein MTBD would be prevented from switching to the low microtubule-binding affinity state. Such a scenario would result in a reduced microtubule dissociation rate (Fig. 1d), and, since microtubule rebinding has been shown to be critical for phosphate release, also a slowed rate of apparent ATP hydrolysis (Fig. 1b, c). However, if this were true, then even in the absence of microtubule binding an MTBD-bound She1 would likely lock the MTBD in the high microtubule affinity state and consequently reduce the rate of ATP hydrolysis, microtubule dissociation rates, and thus velocity. Our findings show that none of these things are true (see Fig. 1c, $k_{basal}$, and Fig. 3e, f). Thus, understanding the relevance of this binding specificity will be the focus of future work.

We found that a chimeric yeast dynein mutant with an MTBD derived from mouse dynein exhibits reduced sensitivity (Fig. 6f–h) and affinity (Fig. 7b, c) for She1. Given that She1 preferentially binds to the MTBD when the latter is in its high microtubule-binding affinity state (see above), one possible explanation for this reduced sensitivity to She1 is that the mouse MTBD—at least in the context of the chimeric motor mutant—is locked in a low (or lower) microtubule affinity state. Consistent with this notion, we found that the GFP–dynein$_{331}$$^{mMTBD}$ chimera exhibited a somewhat lower affinity for microtubules than wild-type GFP–dynein$_{331}$ (Supplementary Fig. 4c). Alternatively, the reduced affinity of She1 for the chimera may simply be a consequence of amino acid substitutions within the MTBD. A comparison of primary sequences between yeast and mouse dynein MTBDs indicates a large number of differences in surface-exposed residues (i.e., those likely contacted by She1; Supplementary Fig. 4b). Specifically, we found there to be 48 surface-exposed residues that are dissimilar, of which 22 are charge

substitutions (i.e., changes that either add, remove, or switch a charge), and 11 are non-polar/polar substitutions. Given the high prevalence of basic residues throughout She1 (isoelectric point of She1 = 10.4), it is possible that the charge substitutions are the basis for disrupted She1–dynein binding in the chimeric mMTBD mutant. As evidence for an electrostatic component to the interaction between She1 and dynein, we previously found that a phosphomimetic She1 mutant (She1$_{5D}$) exhibited a greater effect on dynein motility than wild-type recombinant She1, in spite of the mutant exhibiting a lower microtubule-binding affinity[30]. The majority of residue differences between yeast and mouse MTBDs —including charge substitutions—appear to lie on the right face of the MTBD (Supplementary Fig. 4b). In light of this fact, and that the bulk of the conformational changes induced by nucleotide binding and hydrolysis are clustered on the left face of the MTBD (see above, and Supplementary Fig. 4a), we hypothesize that She1 recognizes a composite binding surface that encompasses both faces of the MTBD. Such a mechanism of binding could account for the apparent high affinity interaction between She1 and dynein ($K_D$ < 0.2 nM; see above), and the high degree of potency with which She1 affects dynein with respect to the only other known molecule that effects dynein similarly: Pac1 (~350-fold difference in half-maximal inhibition[33]). Further study will be required to understand the precise nature of the interaction between She1 and dynein.

The mechanism by which She1 affects dynein-mediated spindle movements is currently unclear. We previously proposed a model in which She1 specifically inhibits dynein activity in the mother cell, which would lead to a relative enhancement in daughter cell-based dynein activity, and consequent daughter cell-directed spindle movements[30]. Although future studies will focus on testing this model, it was unclear from previous work whether the defective spindle neck-cross phenotype in she1Δ cells was due to other, non-dynein-related activities of She1. For instance, She1, which localizes prominently to the bud neck and the mitotic spindle[56, 57], has been implicated in affecting spindle disassembly and kinetochore function, the latter of which may be mediated by Mcm21, a She1 interacting factor and kinetochore component[58] that affects localization of the kinetochore kinase, Ipl1 (homolog of human Aurora B kinase)[59]. Thus, it is possible that the observed defect in spindle neck crossing is attributable to either the bud neck or spindle-localized She1 pools, which presumably do not affect dynein pathway function, as opposed to the astral microtubule-localized She1[57], which is the pool of molecules that likely affects dynein function. Our finding that $dyn1^{mMTBD}$ cells are not further impacted by loss of She1 on spindle neck crossing indicates that it is She1's effect on dynein activity in particular that affects this process in wild-type cells, and that it is likely the astral microtubule-bound population of She1 molecules that are responsible.

## Methods

**Media and strain construction.** Strains are derived from W303, YEF473A[60], or Y2HGold/Y187 (Clontech, catalog number 630489), and are listed in Supplementary Table 1. We transformed yeast strains using the lithium acetate method[61]. Strains carrying mutations or tagged components were constructed by PCR product-mediated transformation[62] or by mating followed by tetrad dissection. Proper tagging and mutagenesis was confirmed by PCR, and in some cases sequencing. Fluorescent tubulin-expressing yeast strains were generated using plasmids and strategies described previously[63]. Yeast synthetic defined (SD) media was obtained from Sunrise Science Products (San Diego, CA).

**Plasmid construction.** A region of dynein corresponding to the CC and MTBDs (CC + MTBD; amino acids 3015–3309; note this fragment is equivalent to the "85:82", "α registry" fragment generated previously[47]) was amplified using forward and reverse primers flanked with SalI and HindIII restriction sites. A bacterial expression vector with mouse dynein$_{CC+MTBD}$ fused to seryl tRNA synthetase[40] was obtained from Addgene (www.addgene.com; plasmid 22393), digested with SalI

and HindIII, and then ligated with the digested yeast dynein$_{CC+MTBD}$ PCR product to generate pSRS:dynein$_{CC+MTBD}$. To generate an N-terminally tagged EGFP variant of this fragment (see Fig. 5a), isothermal assembly was used[64]. PCR products corresponding to EGFP (from pFA6a-GFP(S65T)-TRP[62]) and a portion of the CC + MTBD (amino acids 1–164) were amplified. After amplification, the 5′ end of the EGFP PCR contained 20 nucleotides of sequence identity with NdeI digested pSRS:dynein$_{CC+MTBD}$, and the 5′ and 3′ ends of the CC + MTBD PCR product contained 20 nucleotides of sequence identity with the 3′ end of EGFP, and NdeI digested pSRS:dynein$_{CC+MTBD}$, respectively. After digesting pSRS:dynein$_{CC+MTBD}$ with NdeI (which excises sequence corresponding to amino acids 1–164 of CC + MTBD), the gel purified PCR products and digested vector were assembled in vitro as described[64], yielding pEGFP–SRS:dynein$_{CC+MTBD}$.

To generate pEGFP–SRS:SRS$_{CC}$–dynein$_{MTBD}$ (i.e., in which the native yeast dynein CC is replaced with one from SRS; see Fig. 5a and Supplementary Fig. 3c), a region corresponding to the dynein MTBD (amino acids 3097–3220) was amplified using a forward primer with sequence corresponding to SRS H1 (REVQELKKRLQEVQTERNQVAKR) preceded on the 5′ end by a SalI restriction site, and a reverse primer with sequence corresponding to SRS helix 2 (EEKEALIARGKALGEEAKRLEEALREKEA) preceded on the 5′ end by a HindIII restriction site. Subsequent to amplification, the PCR product was digested with SalI and HindIII, and ligated into pEGFP–SRS:dynein$_{CC+MTBD}$ digested similarly, yielding pEGFP–SRS:SRS$_{CC}$–dynein$_{MTBD}$. A similar construct with the mouse dynein MTBD (amino acids 3279–3401; pEGFP–SRS:SRS$_{CC}$–dynein$_{mMTBD}$) was generated as an intermediate step in constructing the corresponding two-hybrid plasmid (see below). pEGFP–SRS:SRS$_{CC}$–kinesin$_{MTBD}$ (Fig. 5b) was generated similarly, with the only exception being that the forward and reverse primers specifically amplified the kinesin MTBD (amino acids 1–337). Moreover, pEGFP–SRS:linker–kinesin$_{MTBD}$ (i.e., in which the kinesin MTBD is fused to SRS by a flexible linker; see Fig. 5b) was also generated similarly, with the exception being that the forward and reverse primers included nucleotide sequence that encoded flexible linkers (EGKSSGSG on the N-terminus, and KGEGGSSG on the C-terminus).

To generate GAL4-DNA-binding domain (DBD) vectors for the two-hybrid assay, SRS–dynein$_{CC+MTBD}$, SRS–SRS$_{CC}$–dynein$_{MTBD}$, SRS–SRS$_{CC}$–dynein$_{mMTBD}$, and SRS–SRS$_{CC}$–kinesin$_{MTBD}$ were amplified from the respective pEGFP–SRS vectors (described above). After amplification, the 5′ and 3′ ends of each PCR product contained 20 nucleotides of sequence identity with EcoRI and BamHI-digested pGBKT7 (Clontech). After digesting pGBKT7 with EcoRI and BamHI, the gel purified PCR products and digested vector were assembled in vitro as described[64], yielding pGBKT7:SRS–dynein$_{CC+MTBD}$ and pGBKT7: SRS–SRS$_{CC}$–dynein$_{MTBD}$. To construct the GAL4–activation domain (AD)–She1 fusion, a PCR product corresponding to the SHE1 open reading frame was amplified. After amplification, the PCR product contained 20 nucleotides of sequence identity with EcoRI and BamHI-digested pGADT7 (Clontech). After digesting pGADT7 with EcoRI and BamHI, the gel purified PCR product and digested vector were assembled in vitro, yielding pGADT7:SHE1. We found that the ADH1 promoter upstream of GAL4–AD–SHE1 in pGADT7 drove sufficiently high expression of She1 to result in growth arrest (not shown), as has been reported previously for She1-overexpressing cells[30, 65]. Thus, we sought to generate a lower-expressing GAL4–AD–She1 vector. To this end, we PCR amplified 352 nucleotides of genomic DNA sequence upstream of the native yeast SHE1 locus (which presumably contains the native SHE1 promoter, or SHE1p) along with the GAL4–AD–SHE1 open reading frame from pGADT7:SHE1. After amplification, the 5′ and 3′ ends of the two PCR products (SHE1p and GAL4–AD–SHE1) contained 20 nucleotides of sequence identity with each other (i.e., the 3′ end of SHE1p matched the 5′ end of GAL4–AD–SHE1) and with BamHI and NotI-digested pRS315[66] (i.e., the 5′ end of SHE1p matched the BamHI site, and the 3′ end of GAL4–AD–SHE1 matched the NotI site). After digesting pRS315 with BamHI and NotI, the gel purified PCR products and digested vector were assembled in vitro, yielding pRS315:SHE1p:GAL4–AD–SHE1. Yeast cells transformed with this vector did not exhibit any apparent growth defects (Supplementary Fig. 3e; " + HIS" growth). The negative controls (GAL4–DBD–p53 expression vector, pGBKT7–53; and, GAL4–AD–large T antigen-expression vector, pGADT7-T) were obtained from Clontech.

**Protein purification.** We purified She1-HALO as previously described[30], but with minor modifications. Briefly, E. coli BL21 (Rosetta DE3 pLysS) cells transformed with pProEX-HTb-TEV:SHE1-HALO were grown at 37 °C in LB supplemented with 1% glucose, 100 μg/ml carbenicillin, and 34 μg/ml chloramphenicol to OD$_{600}$ 0.4–0.6, shifted to 16 °C for 2 h, then induced with 0.1 mM IPTG for 14–16 h at 16 °C. The cells were harvested, washed with cold water, resuspended in 0.5 volume of cold 2× lysis buffer [1× buffer: 30 mM HEPES pH 7.2, 50 mM potassium acetate, 2 mM magnesium acetate, 0.2 mM EGTA, 10% glycerol, 1 mM DTT, and protease inhibitor tablets (Pierce)] and then lysed by sonication (5 × 30 s pulses) with 1 min on ice between each pulse. The lysate was clarified at 22,000 × g for 20 min, adjusted to 0.01% triton X-100, then incubated with glutathione agarose for 1 h at 4 °C. The resin was then washed three times in wash buffer (30 mM HEPES pH 7.2, 50 mM potassium acetate, 2 mM magnesium acetate, 0.2 mM EGTA, 300 mM KCl, 0.01% Triton X-100, 10% glycerol, 1 mM DTT, protease inhibitor tablets) and twice in TEV digest buffer (10 mM Tris pH 8.0, 150 mM KCl, 0.01% Triton X-100, 10%

glycerol, 1 mM DTT). To fluorescently label She1-HALO, the bead-bound protein was incubated with 6.7 μM HaloTag-TMR ligand (Promega) for 15 min at room temperature. The resin was then washed three more times in TEV digest buffer, then incubated in TEV buffer supplemented with TEV protease for 1 h at 16 °C. The resulting eluate was collected using a centrifugal filter unit (0.1 μm, Millipore), aliquoted, drop frozen in liquid nitrogen and stored at −80 °C. For the ATPase assays, purified She1-HALO was dialyzed against dynein motility buffer (see below) lacking EGTA, but supplemented with 1 mM DTT.

Purification of ZZ–TEV–6His–GFP–3HA–GST–dynein$_{331}$–HALO (under the control of the galactose-inducible promoter, GAL1p) was performed as previously described[33], with minor modifications. Briefly, yeast cultures were grown in YPA supplemented with 2% galactose, harvested, washed with cold water, and then resuspended in a small volume of water. The resuspended cell pellet was drop frozen into liquid nitrogen and then lysed in a coffee grinder (Hamilton Beach). After lysis, 0.25 volume of 4× lysis buffer (1× buffer: 30 mM HEPES, pH 7.2, 50 mM potassium acetate, 2 mM magnesium acetate, 0.2 mM EGTA, 1 mM DTT, 0.1 mM Mg-ATP, 0.5 mM Pefabloc SC, 0.7 μg/ml Pepstatin) was added, and the lysate was clarified at 22,000 × g for 20 min. The supernatant was then bound to IgG sepharose 6 fast flow resin (GE) for 1 h at 4 °C, which was subsequently washed three times in wash buffer (30 mM HEPES, pH 7.2, 50 mM potassium acetate, 2 mM magnesium acetate, 0.2 mM EGTA, 300 mM KCl, 0.005% Triton X-100, 10% glycerol, 1 mM DTT, 0.1 mM Mg-ATP, 0.5 mM Pefabloc SC, 0.7 μg/ml Pepstatin), and twice in TEV buffer (50 mM Tris, pH 8.0, 150 mM potassium acetate, 2 mM magnesium acetate, 1 mM EGTA, 0.005% Triton X-100, 10% glycerol, 1 mM DTT, 0.1 mM Mg-ATP, 0.5 mM Pefabloc SC). Note that for binding experiments involving vanadate (e.g., Fig. 4e), EGTA was excluded from the TEV buffer. To fluorescently label 6His–GFP–GST–3HA–dynein$_{331}$–HALO (for single-molecule analyses), the bead-bound protein was incubated with either 6.7 μM HaloTag-TMR or HaloTag-PEG-biotin ligand (Promega) for 15 min at room temperature. The resin was then washed four more times in TEV digest buffer, then incubated in TEV buffer supplemented with TEV protease for 1 h. Following TEV digest, the bead solution was transferred to a spin column (Millipore) and centrifuged at 20,000 × g for 10 s. The resulting protein solution was aliquoted, flash frozen in liquid nitrogen, and then stored at −80 °C. Protein concentrations were determined by running a dilution series of dynein along with a dilution series of tubulin on a 4–12% SDS–PAGE gel, and then staining the gel with Sypro Red gel stain (Thermo Fisher). Band intensities were quantitatively determined following imaging on a Typhoon gel imaging system (FLA 9500).

Purification of the SRS fusion proteins (dynein$_{CC+MTBD}$, kinesin$_{MTBD}$, SRS$_{CC}$–dynein$_{MTBD}$, and SRS$_{CC}$–kinesin$_{MTBD}$; Fig. 5a, b) were performed essentially as described[40, 47] with minor modifications. E. coli BL21 cells transformed with the appropriate vector (described above in Plasmid construction) were grown at 30–37 °C in LB, 30 μg/ml kanamycin and 34 μg/ml chloramphenicol to OD$_{600}$ 0.4–0.6, shifted to 16 °C for 2 h, then induced with 0.1 mM IPTG for 14–16 h at 16 °C. The cells were harvested, washed with cold water, resuspended in cold lysis buffer (30 mM HEPES pH 8.0, 50 mM potassium acetate, 2 mM magnesium acetate, 10% glycerol, 10 mM imidazole, 5 mM beta-mercaptoethanol, protease inhibitor tablets) and then lysed by sonication (5 × 30 s pulses) with 1 min on ice between each pulse. The lysate was clarified at 22,000 × g for 20 min, then incubated with Ni-NTA agarose (Qiagen) for 1 h at 4 °C. The resin was then washed three times in lysis buffer, after which the resin was transferred to a disposable column, and the protein was eluted with elution buffer (30 mM HEPES pH 8.0, 50 mM potassium acetate, 2 mM magnesium acetate, 10% glycerol, 200 mM imidazole, 5 mM beta-mercaptoethanol). Peak fractions were pooled and applied to a Superdex 200 (10/300) gel filtration column (using an AKTA fast protein liquid chromatography system) equilibrated in lysis buffer. Peak gel filtration fractions were pooled, concentrated (to between 47 and 89 μM) in a centrifugal filter device (Amicon Ultra-2 ml, Millipore), aliquoted, and drop frozen in liquid nitrogen. We noted that we were able to obtain higher SRS fusion protein concentrations in pH 8.0 than in pH 7.2 buffer. We ensured that these pH differences between protein purification buffers were carefully controlled for in the binding assays described below (see Microtubule recruitment assays, below).

**Single and ensemble molecule motility assays.** The single-molecule motility assay was performed as previously described[30] with minor modifications. Briefly, flow chambers constructed using slides and plasma cleaned and silanized coverslips attached with double-sided adhesive tape were coated with anti-tubulin antibody (8 μg/ml, YL1/2; Accurate Chemical & Scientific Corporation) then blocked with a mixture of 1% Pluronic F-127 (Fisher Scientific) and 1 mg/ml κ-casein. Taxol-stabilized microtubules (either digested with subtilisin, as described below in Microtubule recruitment assays, or undigested) assembled from unlabeled and HiLyte647-labeled porcine tubulin (10:1 ratio; Cytoskeleton) were introduced into the chamber. Following a 5–10 min incubation, the chamber was washed with dynein lysis buffer supplemented with 20 μM taxol, at which point She1-488 was added to the chamber. After a 5-min incubation, 6His–GST–dynein$_{331}$–TMR diluted (~10 pM) in motility buffer (30 mM HEPES, pH 7.2, 50 mM potassium acetate, 2 mM magnesium acetate, 1 mM EGTA, 10% glycerol) supplemented with 1 mM DTT, 20 μM taxol, 1 mM Mg-ATP, 0.05% Pluronic F-127, and an oxygen-scavenging system (1.5% glucose, 1 U/μl glucose oxidase, 125 U/μl catalase) was added. TIRFM images were collected using a 1.49 NA 100× TIRF objective on a

Nikon Ti-E inverted microscope equipped with a Ti-S-E motorized stage, piezo Z-control (Physik Instrumente), and an iXon × 3 DU897 cooled EM-CCD camera (Andor). 488 nm, 561 nm, and 640 nm lasers (Coherent) were used along with a multi-pass quad filter cube set (C-TIRF for 405/488/561/638 nm; Chroma) and emission filters mounted in a filter wheel (525/50 nm, 600/50 nm, and 700/75 nm; Chroma) to image She1-488, 6His–GST–dynein$_{331}$–TMR, and HiLyte647-microtubules, respectively. We acquired images at 2 s intervals for 10 min. Velocity and run length values were determined from kymographs generated using the MultipleKymograph plugin for ImageJ (http://www.embl.de/eamnet/html/body_kymograph.html). Run length and dwell time for individual runs were determined by fitting with cumulative distribution functions (see Supplementary Figs 2 and 5), as previously described[46].

For super resolution stepping analysis, high temporal resolution (~10 fps) movies were acquired of Quantum dot-labeled dynein molecules as previously described[46]. Briefly, low concentrations (~10 pM) of chamber-immobilized microtubule-bound 6His–GST–dynein$_{331}$–PEG–biotin molecules were incubated with 100 nM 525 Qdot streptavidin (Thermo Fisher) under conditions that yield monovalent Qdot attachment[46] (note that for experiments with She1, the chambers were pre-incubated with the indicated concentrations of She1 prior to motor addition). Subsequently, the chambers were washed sequentially with motility buffer (with, or without She1, as indicated), and then motility buffer supplemented with 0.05% Pluronic F-127, the oxygen-scavenging system (see above), and either 1 mM Mg-ATP, or 1 µM Mg-ATP (see figures and/or figure legends), and the indicated concentration of She1. For low (1 µM) ATP conditions, the motility buffer was further supplemented with an ATP regenerating system (1% pyruvate kinase and 10 mM phosphenolpyruvate). TIRFM images were recorded every 100 ms, and fluorescent spots were fitted with a 2D Gaussian function to precisely localize their position as previously described[67]. We found that Qdot 525 provided us with the highest signal-to-noise images; however, this fluorophore exhibits overlapping excitation and emission profiles with the GFP near the N-terminus of GST–dynein$_{331}$ (i.e., 6His–GFP–3HA–GST–dynein$_{331}$–HALO; Supplementary Fig. 1a). With our imaging conditions, the GFP photobleached quite rapidly with respect to the photostable Qdot. Specifically, we found that there was a 99.5% probability that GFP photobleached within 388 frames (38.8 s) of first appearing (Supplementary Fig. 1b). This is in striking contrast to the Qdot$^{525}$, which was extremely photostable. Thus, to ensure our particle detection algorithm was tracking Qdot-labeled dynein (i.e., not GFP), the first 400 frames of each processive run were discarded. Steps were detected from the displacement records using a custom-written Mathematica (Wolfram Research) program (available upon request). Steps were assigned only if the dwells before and after contained at least three frames[68].

For microtubule gliding (i.e., ensemble motor motility) assays (Fig. 6e, top), flow chambers were coated with anti-His$_6$ (Roche) antibody for 5 min, and then blocked as above. 6His–GST–dynein$_{331}$ (wild-type or chimera; 5 µg/ml) was subsequently introduced into the chamber, incubated for 2 min, and then washed with one chamber volume of motility buffer. The chamber was then washed with motility buffer supplemented with the oxygen-scavenging system (see above), 1 mM Mg-ATP, and HiLyte647-microtubules (125 nM), after which TIRFM images were collected every 5 s. For experiments in which She1 was included, the motility mix with microtubules was pre-incubated with 10 nM She1-HALO for 10 min prior to its addition to the chamber. Velocity values were determined from kymographs generated as described above.

**Dynein ATPase assays.** Basal and microtubule-stimulated ATPase activities were determined using the EnzChek phosphate assay kit (Life Technologies). Assays were performed in motility buffer (see above) supplemented with 2 mM Mg ATP, with 0–2 µM taxol-stabilized microtubules, 5 nM 6His–GST–dynein$_{331}$, and in the absence or presence of 200 nM She1. Reactions were initiated with the addition of dynein, and the absorbance at 360 nm was monitored by a spectrophotometer for 10–20 min. Background phosphate release levels (presumably from microtubules) for each reaction were measured for 5 min before addition of dynein to account for any variation as a consequence of differing microtubule concentrations, and were subtracted out from each data point. $K_{m(MT)}$, $k_{basal}$, and $k_{cat}$ were determined from fitting the data to Eq. (1), as previously described[34], where $k_{obs}$ and $k_{basal}$ are the observed and basal ATPase rates, and $x$ is the concentration of tubulin that used to generate microtubules for a given data point:

$$k_{obs} = \left( \frac{x(k_{cat} - k_{basal})}{(K_{m(MT)} + x) + k_{basal}} \right)^2 \qquad (1)$$

**Microtubule recruitment assays.** Taxol-stabilized microtubules were digested with a freshly dissolved preparation of 1–2 mg/ml subtilisin (Sigma; from a stock solution of 5 mg/ml) for 60–75 min at 37 °C prior to each binding assay. Chambers were prepared as described above (Single and ensemble molecule motility assays). After microtubules were adhered to the cover glass, mixtures of She1-TMR and dynein fragments (as described throughout the text and in figure legends) were flowed into the chambers for 5 min, after which the chambers were washed with motility buffer (see above), and immediately imaged. Buffer conditions for a given binding experiment were kept constant to ensure that buffer conditions (e.g., salt

concentration, etc) were not factors in the apparent degree of microtubule recruitment. For the SRS–MTBD/She1 recruitment assays, the pH of the reaction mixtures was kept constant between samples by mixing motility buffers (see above) that differed only in their pH: pH 6.7 and pH 8.0. The final reaction buffer consisted of 61% motility buffer pH 6.7, and 39% motility buffer pH 8.0 (resulting in a final pH of 7.4). For experiments in which relative She1 microtubule recruitment was quantitatively compared (e.g., Figs. 4e and 5f), imaging conditions were kept constant (i.e., laser power and camera exposures). Moreover, to control for differences in labeling efficiencies of She1-HALO (with the HALO-TMR ligand), protein from a given preparation was only compared to itself (i.e., protein from different preps were never used for a given experimental replicate). Quantitation of the recruitment assays was performed using ImageJ software (National Institutes of Health). Fluorescence intensities in the red (She1-TMR) and green (dynein or kinesin fragments) channels were measured along microtubules ("signal"; determined from HiLyte-647-microtubule fluorescence), and adjacent to microtubules ("background"). Mean corrected pixel intensity was determined by subtracting background from signal. To correct for differential microtubule-binding affinity of the various protein fragments (e.g., dynein$_{MTBD}$ vs SRS$_{CC}$–dynein$_{MTBD}$; Fig. 5f), fluorescence intensity values in the green channel (GFP) were used in place of concentration. Binding curves and curve fitting for dissociation constants (where appropriate) were generated using GraphPad Prism.

**Live cell imaging experiments.** For the single time point spindle position assay, the percentage of cells with a misoriented anaphase spindle was determined after growth overnight (12–16 h) at a low temperature (16 °C), as previously described[49, 69, 70]. A single z-stack of wide-field fluorescence images was acquired for mRuby2-Tub1. For the spindle oscillation assay (Fig. 7g, h and Supplementary Fig. 7c), cells were arrested with HU for 2.5 h, and then mounted on agarose pads containing HU for fluorescence microscopy. GFP-labeled microtubules (GFP-Tub1) were imaged every 10 s for 10 min. To image dynein localization in live GAL1p:SHE1 cells (Fig. 7e, f, and Supplementary Fig. 6), cells were grown as indicated in figure legends, and mounted on agarose pads. Images were collected on a Nikon Ti-E microscope equipped with a 1.49 NA 100× TIRF objective, a Ti-S-E motorized stage, piezo Z-control (Physik Instrumente), an iXon DU888 cooled EM-CCD camera (Andor), and a spinning disc confocal scanner unit (CSUX1; Yokogawa) with an emission filter wheel (ET480/40 M for mTurquoise2, ET525/50 M for GFP, ET520/40 M for YFP, and ET632/60 M for mRuby2; Chroma). A total of 445 nm, 488 nm, 515 nm, and 561 nm lasers (housed in a LU-NV laser unit equipped with AOTF control; Nikon) were used to excite mTurquoise2, GFP, YFP and mRuby2, respectively. The microscope was controlled with NIS Elements software (Nikon). Image analysis was performed using ImageJ software (National Institutes of Health). Plus end and SPB foci were identified in two-color movies and scored accordingly. Specifically, plus end molecules were recognized as those foci that localized to the distal tips of dynamic microtubules (identified via mTurquoise2-Tub1 or mRuby2-Tub1 imaging), whereas spindle pole body (SPB)-associated molecules were recognized as those foci that localized to one of the spindle poles. Cortical molecules were identified as those foci not associated with an astral microtubule plus end that remained stationary at the cell cortex for at least three frames, whereas cytoplasmic foci were identified as those dynamic foci not meeting the criteria described for any of the above described categories (i.e., not associating with astral microtubules, or SPBs).

**Yeast two-hybrid assay.** For each assay, an equivalent number of yeast cells containing plasmids expressing a GAL4–DBD and transcriptional AD ("AD") fusions were spotted onto histidine-containing plates ("+HIS"; as control), or selective media lacking histidine ("−HIS"), the latter of which contained 5 mM 3-amino-1,2,4-triazole (to reduce background growth due to autoactivation by GAL4–DBD–SRS$_{CC}$–dynein$_{MTBD}$ bait; not shown). Both the +HIS and −HIS plates lacked tryptophan and leucine in order to select for cells containing both plasmids.

**Statistical analyses.** P-values were calculated using a two-tailed unpaired t test. For box-whisker plots, whiskers define the range, boxes encompass 25th to 75th quartiles, lines depict the medians, and circles depict outlier values (defined as values greater than (upper quartile + 1.5 × interquartile distance), or less than (lower quartile − 1.5 × interquartile distance)).

**Data availability.** All datasets generated during the course of this study are available from the corresponding author upon reasonable request.

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

## Acknowledgements

We are grateful to Roderick Lammers for help with R, to Olve Peersen for assistance with homology modeling, sharing equipment and valuable discussions, and members of the Markus and DeLuca laboratories for valuable discussions. We are also grateful to Samara Reck-Peterson and Nathan Derr for sharing yeast strains. This work was funded by the Muscular Dystrophy Association (376387 to S.M.M.), the NIH/NIGMS (GM118492 to S. M.M. and GM119728 to T.J.S.), and the W.M. Keck Foundation (to T.J.S.).

## Author contributions

S.M.M. designed the study. S.M.M. and K.H.E. generated the reagents, performed experiments, and analyzed the single-molecule and binding data. L.G.L. performed the ATPase assays, and analyzed the data. T.M. and T.J.S. performed the high resolution stepping analysis. S.M.M., M.G.M. and K.H.E. acquired and analyzed the spindle oscillation and orientation data. S.M.M. performed the two-hybrid assay, and wrote the manuscript.

## Additional information

**Competing interests:** The authors declare no competing financial interests.

