## [Peer Review File · Nature Communications]

Reviewers' comments:

Reviewer #1 (Remarks to the Author):

The manuscript by Ecklund et al. reports the first molecular mechanism by which a microtubule (MT)-associated protein (MAP) controls the activity of a MT-associated motor. The experiments are outstanding and the provided insights into the regulation of cytoplasmic dynein by the MAP, She1, are seminal for the dynein community. The manuscript is written with care and of high interest for the cell biology and cytoskeletal motor communities. I feel that it will be a timely and well-cited contribution. I therefore recommend publication, subject to the following minor changes:

1. Last paragraph of introduction, "We confirm the She1-MTBD interaction by generating a dynein mutant that exhibits a reduced binding affinity for dynein and is less sensitive to She1 effects in vitro and in vivo". I believe the authors meant to write "...by generating a dynein mutant that exhibits a reduced binding affinity for She1 and is less sensitive to She1 effects in vitro and in vivo".
2. Figure 3d: I recommend to indicate directly in the figure panel that the given numbers correspond to the applied She1 concentrations.

Reviewer #2 (Remarks to the Author):

The manuscript by Ecklund et al. reports that one of the microtubule-associated proteins (MAPs), She1, directly binds to yeast cytoplasmic dynein and modulates dynein motility and ATPase in a nucleotide-state dependent manner. By using several experimental systems including single molecule measurements, the authors characterized the molecular basis by which She1 affects dynein motility and made several intriguing findings that will significantly advance the research field of MAPs and dyneins. For example, the authors showed that She1 specifically reduced the ATPase activity and stepping rates of dynein, and that She1 simultaneously interacted with microtubules and microtubule-binding domain of dynein. The authors also succeeded to show that She1 recognized a structural feature of dynein and preferentially bound to apo conformation of microtubule binding domain of dynein by single molecule measurements.

One worry was that this work draws conclusions from an artificially dimerized motor complex, which may not necessarily represent the conformational organization and regulation of the native dynein complex. The artificial dimerization may result in a non-physiological mechanism of She1 regulation. However, the authors carried out the She1-overexpression experiments combined with dynein mutants in yeast cells and clearly showed the consistency of results obtained by in vitro and in vivo experiments.

Therefore, the referee thinks that this is a carefully done and well controlled study and findings and the experimental system presented in this manuscript are highly original and of considerable interest. The relevance of the questions asked surely justifies publication in Nature Communications.

Minor points.

Although the publisher may take care of the drawing, in the labels of each Figures, the lower-case characters, such as Fig. 1a, b are used but in text the upper-case characters, such as Fig1A, B are used. Please unify them.

Reviewer #3 (Remarks to the Author):

Title: She1 affects dynein by interactions with the microtubule and the dynein microtubule-binding

domain

Summary

The manuscript by Ecklund et al. reports *in vitro* molecular studies that delineate the mechanism of She1 inhibition of the dynein motor protein on microtubules. The authors have previously identified the yeast specific microtubule binding protein, She1, interacts with dynein and stalls its motility *in vitro*. In current manuscript, the authors did further detailed characterization of She1 effects on dynein activity using mainly *in vitro* approaches coupled with *in vivo* analysis. The authors demonstrated that She1 directly binds to the dynein MTBD, providing the first such example of an allosteric regulatory protein interacting with this critical region of the dynein motor. She1 dose dependently decreases the stepping rate of dynein, which mimics stepping behavior under low ATP concentration, while mildly increasing backward or longer stepping size as well. She1 inhibits microtubule dependent activation of ATPase activity of dynein, increases dwell time of dynein on microtubule which indicates She1 reduces off rate of dynein, and further, She1 specifically interacts with dynein in apo-state, all pointing to the notion that She1 inhibits dynein motility by locking in the conformation that binds strongly to microtubule. This inhibitory effect of She1 on dynein is dependent on its binding to microtubule, as well as direct interaction with microtubule binding domain of dynein. Finally, the authors demonstrated in budding yeast that the disruption of dynein binding to she1 greatly reduces crossing of spindle through bud neck to the daughter cell in *kar9Δ* background, but doesn't have any synthetic effect when combined with *she1Δ* suggesting that this interaction is likely the cause of the *she1Δ* defect.

The data presented in this manuscript appear to be of high quality and carefully measured, and the biochemical characterization of She1 is beneficial to understand cytoskeletal organization of budding yeast. The biggest drawback of the study is that She1 is a yeast specific protein and the dynein chimera containing the mouse MTBD was not compatible with She1 inhibition, suggesting the mechanism of She1 inhibition is not conserved in metazoans. It is therefore not entirely clear how this study is applicable to other model systems beyond yeast. Nonetheless, the paradigm of MAP regulation of molecular motor transport reported here provides very interesting new insight that should be of high interest to the cytoskeletal community and could stimulate the search for molecules in metazoans that perform similar functions. We support publication after the authors address the following concerns:

Major concerns:

1. It is an interesting observation that the difference in between MTBD in mouse dynein and that of budding yeast are mostly clustered on one side of MTBD (called right side in the manuscript), which the authors speculate as a binding surface for She1 interaction. At the same time, the authors also speculate that the binding surface a composite that encompasses the region in MTBD that undergoes conformational changes (CC and H1), that are on the opposite surface. Since She1 has very high binding affinity, the authors could attempt cross-linking experiments to identify which specific residues are involved in She1-dynein interaction. Alternatively, site-directed mutagenesis on non-conserved charged residues on the "right side" of MTBD to examine if it abolishes the binding would be beneficial to strengthen the manuscript and provide more direct insight into the mechanism of the interaction. Such experiments would make the manuscript more impactful but are not a requirement for publication.

2. What is the stalk registry of the SRS constructs used (+ β , α , - β , Kon et al. 2009, Gibbons et al 2005)? Changing the registry of the coiled-coil, as has been done previously, seems like a better way to examine the interaction of She1 with different conformations of the MTBD. Could the authors try the α and + β registries to examine the differences in She1 binding as this would be more related to the native conformations of the dynein MTBD than in their SRScc mutants.

5. The authors speculate that the reason why She1-bound dynein can still move processively along subtilisin-treated MT is that only one motor domain of dynein is occupied by She1. This could be further addressed by observing step-wise photobleaching of both dynein (or SRS-MTBD) and She1 on subtilisin MTs to further support this stoichiometry.

6. Can the mMTBD rescue spindle the positioning error by She1 OE since the chimera is not sensitive to She1?

7. No statistics are provided about the percent of dynein molecules that co-migrate with She1 in Fig. 3f. How rare is this observation?

Minor concerns:

1. The histograms in Fig. S1C appear like they may be switched around. Should the red and blue graphs be switched?

2. Fig.3d right panels: Qualitatively, the kymographs look somewhat different in the presence of 10 nM She1 versus its absence, that there appear to be more short-lived dynein runs, but quantitatively there is no significant difference (Fig3e and f). Are these kymographs representative?

3. Using the term "mutant" to describe the MTBD swap is misleading. We suggest replacing with "chimeric"

Detailed Response to Referee #1

We thank the referee for valuable comments that helped improve our manuscript.

“The manuscript by Ecklund et al. reports the first molecular mechanism by which a microtubule (MT)-associated protein (MAP) controls the activity of a MT-associated motor. The experiments are outstanding and the provided insights into the regulation of cytoplasmic dynein by the MAP, She1, are seminal for the dynein community. The manuscript is written with care and of high interest for the cell biology and cytoskeletal motor communities. I feel that it will be a timely and well-cited contribution. I therefore recommend publication, subject to the following minor changes:

1. Last paragraph of introduction, “We confirm the She1-MTBD interaction by generating a dynein mutant that exhibits a reduced binding affinity for dynein and is less sensitive to She1 effects in vitro and in vivo”. I believe the authors meant to write “...by generating a dynein mutant that exhibits a reduced binding affinity for She1 and is less sensitive to She1 effects in vitro and in vivo”.

We thank the reviewer for catching this mistake, which has now been corrected.

2. Figure 3d: I recommend to indicate directly in the figure panel that the given numbers correspond to the applied She1 concentrations.

We thank the reviewer for suggesting this addition to Figure 3d, which has now been updated.

Detailed Response to Referee #2

We thank the referee for valuable comments that helped improve our manuscript.

“The manuscript by Ecklund et al. reports that one of the microtubule-associated proteins (MAPs), She1, directly binds to yeast cytoplasmic dynein and modulates dynein motility and ATPase in a nucleotide-state dependent manner. By using several experimental systems including single molecule measurements, the authors characterized the molecular basis by which She1 affects dynein motility and made several intriguing findings that will significantly advance the research field of MAPs and dyneins. For example, the authors showed that She1 specifically reduced the ATPase activity and stepping rates of dynein, and that She1 simultaneously interacted with microtubules and microtubule-binding domain of dynein. The authors also succeeded to show that She1 recognized a structural feature of dynein and preferentially bound to apo conformation of microtubule binding domain of dynein by single molecule measurements.

One worry was that this work draws conclusions from an artificially dimerized motor complex, which may not necessarily represent the conformational organization and regulation of the native dynein complex. The artificial dimerization may result in a non-physiological mechanism of She1 regulation. However, the authors carried out the She1-overexpression experiments combined with dynein mutants in yeast cells and clearly showed the consistency of results obtained by in vitro and in vivo experiments. Therefore, the referee thinks that this is a carefully done and well controlled study and findings and the experimental system presented in this manuscript are highly original and of considerable interest. The relevance of the questions asked surely justifies publication in Nature Communications.”

1. *“Although the publisher may take care of the drawing, in the labels of each Figures, the lower-case characters, such as Fig. 1a, b are used but in text the upper-case characters, such as Fig1A, B are used. Please unify them.”*

We have updated the figure call-outs in the manuscript to comply with Nature press formatting, such that lower-case letters are used throughout. We thank the referee for catching this mistake.

Detailed Response to Reviewer #3

We thank the referee for valuable comments that helped improve our manuscript.

“The manuscript by Ecklund et al. reports in vitro molecular studies that delineate the mechanism of She1 inhibition of the dynein motor protein on microtubules. The authors have previously identified the yeast specific microtubule binding protein, She1, interacts with dynein and stalls its motility in vitro. In current manuscript, the authors did further detailed characterization of She1 effects on dynein activity using mainly in vitro approaches coupled with in vivo analysis. The authors demonstrated that She1 directly binds to the dynein MTBD, providing the first such example of an allosteric regulatory protein interacting with this critical region of the dynein motor. She1 dose dependently decreases the stepping rate of dynein, which mimics stepping behavior under low ATP concentration, while mildly increasing backward or longer stepping size as well. She1 inhibits microtubule dependent activation of ATPase activity of dynein, increases dwell time of dynein on microtubule which indicates She1 reduces off rate of dynein, and further, She1 specifically interacts with dynein in apo-state, all pointing to the notion that She1 inhibits dynein motility by locking in the conformation that binds strongly to microtubule. This inhibitory effect of She1 on dynein is dependent on its binding to microtubule, as well as direct interaction with microtubule binding domain of dynein. Finally, the authors demonstrated in budding yeast that the disruption of dynein binding to she1 greatly reduces crossing of spindle through bud neck to the daughter cell in kar9Δ background, but doesn’t have any synthetic effect when combined with she1Δ suggesting that this interaction is likely the cause of the she1Δ defect. The data presented in this manuscript appear to be of high quality and carefully measured, and the biochemical characterization of She1 is beneficial to understand cytoskeletal organization of budding yeast. The biggest drawback of the study is that She1 is a yeast specific protein and the dynein chimera containing the mouse MTBD was not compatible with She1 inhibition, suggesting the mechanism of She1 inhibition is not conserved in metazoans. It is therefore not entirely clear how this study is applicable to other model systems beyond yeast. Nonetheless, the paradigm of MAP regulation of molecular motor transport reported here provides very interesting new insight that should be of high interest to the cytoskeletal community and could stimulate the search for molecules in metazoans that perform similar functions. We support publication after the authors address the following concerns:”

Major Comments

1. *“It is an interesting observation that the difference in between MTBD in mouse dynein and that of budding yeast are mostly clustered on one side of MTBD (called right side in the manuscript), which the authors speculate as a binding surface for She1 interaction. At the same time, the authors also speculate that the binding surface a composite that encompasses the region in MTBD that undergoes conformational changes (CC and H1), that are on the opposite surface. Since She1 has very high binding affinity, the authors could attempt cross-linking experiments to identify which specific residues are involved in She1-dynein interaction. Alternatively, site-directed mutagenesis on non-conserved charged residues on the “right side” of MTBD to examine if it abolishes the binding would be beneficial to strengthen the manuscript and provide more direct insight into the mechanism of the interaction. Such experiments would make the manuscript more impactful but are not a requirement for publication.”*

The referee’s suggestion to perform cross-link/mass-spec analysis to refine the binding

surfaces between She1 and dynein could indeed, if successful, be a highly informative strategy that would reveal additional insight into the precise residues within the dynein MTBD that are contacting She1, and vice versa. However, this is a laborious and challenging experiment that we are not currently suited to perform in my lab. Such an experiment is by no means trivial, and does not necessarily have a high likelihood of success.

Although a site-directed mutagenesis strategy, as suggested by the referee, would also be potentially informative, it would also be complicated in that mutating residues throughout the 124 amino acid MTBD could potentially result in detrimental mutations that disrupt dynein structure/function, and not just She1-binding *per se*. Given that our current data suggest that She1 might in fact contact residues on more than just the “right” face, but both faces, it is currently unclear how many residues should be targeted for site-directed mutagenesis, and moreover which ones. Finally, I’d like to point out that we have in fact narrowed down the site of interaction to a very small region of dynein (~3% of the total residues), which we believe is highly revelatory of the mechanism of their interaction.

These are indeed great experimental suggestions that we intend to address in a future manuscript in which we will further refine the binding surfaces between She1 and dynein. We already have in mind some strategies that would directly address these points, but they are likely a multi-year endeavor. For these reasons, we feel these experiments are beyond the scope of our current manuscript.

2. “What is the stalk registry of the SRS constructs used (+ β , α , - β , Kon *et al.* 2009, Gibbons *et al.* 2005)? Changing the registry of the coiled-coil, as has been done previously, seems like a better way to examine the interaction of She1 with different conformations of the MTBD. Could the authors try the α and + β registries to examine the differences in She1 binding as this would be more related to the native conformations of the dynein MTBD than in their SRScc mutants.”

Although we can posit the stalk registry of the SRS-dynein_{CC+MTBD} fusion based on previous work with mouse dynein (α registry; similar to the “85:82” construct used by Carter *et al.*, 2008), the same registry terminology would not be applicable to our SRS-SRS_{CC}-dynein_{MTBD} fusion since this construct lacks dynein coiled-coil. That being said, our goal in designing this construct was for the MTBD to be in a high MT-binding affinity state, much like the SRS-dynein_{CC+MTBD} fusion. To our surprise, this smaller fragment bound MTs with higher affinity than the SRS-dynein_{CC+MTBD} fusion.

For several reasons, we think our experiments with the entire motor domain combined with our experiments with the SRS-dynein_{CC+MTBD} and SRS-SRS_{CC}-dynein_{MTBD} constructs provide a less ambiguous system to determine which state She1 preferentially binds than the two SRS fusions suggested by the referee.

- a) Firstly, we have performed preliminary characterization of a + β version of the SRS-dynein_{CC+MTBD} fusion derived from yeast dynein (*i.e.*, similar to the “89:82” construct used by Carter *et al.*, 2008), and we were surprised to see little to no difference in MT binding affinity between this construct and the α variant (not shown). Although it’s unclear why we observed no difference in MT binding affinity between these two variants, we hypothesize that it may be due to differences between mouse and yeast dynein. Without high-resolution structural information (*e.g.*, crystal structure), there is no reason to assume *a priori* that the SRS fusion with the native dynein CC fused to the globular region of SRS is in a more native state than our minimal MTBD fragment fused to the SRS coiled-coil (SRS-SRS_{CC}-dynein_{MTBD}). In fact, given that we observed no difference in binding affinity between the + β and α variants indicates that we cannot rely on the lengths of CC1 and CC2 (*e.g.*, 85:82, or 89:82) as a

predictor of the conformation of the yeast dynein MTBD. We instead found the empirically measured MT binding affinity to be a more accurate predictor of MTBD conformation (*i.e.*, high vs. low MT-binding affinity states).

- b) Secondly, comparing the minimal MTBD fragment (SRS-SRS_{CC}-dynein_{MTBD}) to the longer, lower MT-affinity SRS-dynein_{CC+MTBD} fusion (see Fig. S3d) permitted us to rule out any contribution of the dynein coiled-coil to She1's preferential binding. The fact that the shorter fragment happened to bind MTs with higher affinity was serendipitous, and permitted us to conclude that it was indeed in a higher MT-binding affinity conformation than the longer construct.
- c) Lastly, by performing She1 binding (recruitment) assays with the entire motor domain fragments (*i.e.*, dynein₃₃₁, as in Fig. 4) in the absence or presence of ATP + vanadate (which locks the CC+MTBD in either a high or low MT-binding affinity state), we were able to conclude that the She1 preferential binding was indeed a consequence of it recognizing distinct conformational states of the dynein MTBD in its native motor domain context (*i.e.*, more so than the SRS fusions).

We have updated the manuscript to indicate the predicted registry of the SRS-dynein_{CC+MTBD} fusion. We thank the referee for the suggestion.

3. “The authors speculate that the reason why She1-bound dynein can still move processively along subtilisin-treated MT is that only one motor domain of dynein is occupied by She1. This could be further addressed by observing step-wise photobleaching of both dynein (or SRS-MTBD) and She1 on subtilisin MTs to further support this stoichiometry.”

To clarify, our hypothesis is that She1 may only bind to one motor head during brief periods of backward stepping along control, undigested microtubules. We do not mean to imply that this is representative of the stoichiometry on undigested (non-subtilisin-treated) microtubules when She1 is present at nanomolar concentrations. We apologize for the confusion. Given the predicted high affinity of She1 for dynein, at a sufficiently high density of microtubule-bound She1, we would in fact predict that both motor heads would be bound by She1 more often than not. We have modified the text to clarify this point as follows:

*“She1 is the first molecule identified to date that has the capacity to alter dynein stepping behavior (*i.e.*, increases the frequency of large and backward steps; Fig. 2g and h). Although the reasons for this are unclear, we hypothesize that these changes in stepping behavior are a consequence of one of the motor heads within a dimer becoming unbound from She1 for brief periods of time. In such a scenario, one motor head unbinds from microtubule-bound She1 and steps forward. Given the lower likelihood of the lagging She1-bound head unbinding from the microtubule (due to reduced dissociation rates; see Fig. 1d), the leading She1-unbound head in this scenario will unbind from the microtubule and consequently steps backward.”*

Moreover, we would predict that the reason that She1-bound dynein can still move processively along subtilisin-treated microtubules is that She1 binds to a surface of dynein that does not interfere with stepping (*i.e.*, She1 does not likely bind between the MTBD and the MT), or with the conformational changes that dynein must undergo to walk processively. We don't think this has anything to do with only one motor domain being bound by She1. In fact, given that each dynein dimer has two She1-binding sites (*i.e.*, two MTBDs), we would predict that each dynein walking along subtilisin microtubules will be bound by either 1 or 2 She1 molecules.

Unfortunately, observations of single molecules of She1 moving with dynein along subtilisin-treated microtubules were quite rare (see below). We needed to acquire several movies to observe a few events. This is likely due to the low concentrations of dynein we needed to use in order to (1) observe single molecule events (picomolar quantities), and (2) to avoid excessive crowding of the motors on the microtubules, which could potentially affect

motor motility. The rarity of such events would complicate a stepwise photobleaching approach to determine the dynein-She1 stoichiometry. We are hoping that future endeavors using single particle EM techniques (e.g., cryo-EM, or negative stain EM) will permit us to unambiguously determine the nature of the interaction between dynein and She1, and also permit an estimation of their stoichiometry.

4. “Can the mMTBD rescue spindle the positioning error by She1 OE since the chimera is not sensitive to She1?”

We have now experimentally addressed this question using a single time-point spindle orientation assay (see Supplementary Fig. 7a). As expected based on the mislocalization of the dynein^{mMTBD} mutant upon She1 overexpression (as described on page 18 of our new manuscript), spindle orientation was equally compromised in *DYN1*, *dyn1Δ*, and *dyn1^{mMTBD}* cells upon She1 overexpression. We also noted that She1 overexpression resulted in a spindle misorientation phenotype that was more severe than deleting Dyn1 ($p \leq 0.015$), suggesting that She1 overexpression disrupts both pathways used by yeast for spindle orientation (i.e., dynein and Kar9). We thank the referee for the suggestion.

5. “No statistics are provided about the percent of dynein molecules that co-migrate with She1 in Fig. 3f. How rare is this observation?”

For reasons discussed above, these events were indeed rare. In total, we observed only 14 such events (out of several hundred motile dynein molecules). We have updated the manuscript to reflect this. We thank the referee for pointing this out.

Minor Points

1. “The histograms in Fig. S1C appear like they may be switched around. Should the red and blue graphs be switched?”

The graphs are correct as depicted. Note that the stepping assay for our ‘minus She1’ dataset was performed in very limiting ATP (1 μ M). This leads to a very low stepping rate (i.e., long dwell times) that is equivalent to the stepping rate of dynein in the presence of saturating ATP (1 mM) and 25 nM She1 (brown histogram), but less than the stepping rate of dynein in the presence of only 10 nM She1 (red histogram).

2. “Fig.3d right panels: Qualitatively, the kymographs look somewhat different in the presence of 10 nM She1 versus its absence, that there appear to be more short-lived dynein runs, but quantitatively there is no significant difference (Fig3e and f). Are these kymographs representative?”

We thank the referee for pointing this out. We have changed the kymograph in Fig. 3d (“-E-hooks,” 0 nM She1) to one that is more representative of the data.

3. “Using the term “mutant” to describe the MTBD swap is misleading. We suggest replacing with “chimeric.”

Given that the molecule in question is distinct from the native yeast dynein, we think that either “mutant” or “chimeric” would accurately describe this motor. For this reason, we would prefer to leave the terminology as is.

REVIEWERS' COMMENTS:

Reviewer #3 (Remarks to the Author):

The revised manuscript by Ecklund et al. has addressed the concerns raised in the previous round of review and I support publication after the authors consider the following minor points.

1. "Using the term "mutant" to describe the MTBD swap is misleading. We suggest replacing with "chimeric."

Given that the molecule in question is distinct from the native yeast dynein, we think that either "mutant" or "chimeric" would accurately describe this motor. For this reason, we would prefer to leave the terminology as is. **

I still think the use of the term 'mutant' will be misleading for many readers. Most of the time in this field 'mutant' is used to describe a protein with one, or a small set, of changed amino-acids. In this case, the authors are referring to proteins in which entire domains, comprising dozens of amino acids, have been swapped between species.

1. "The histograms in Fig. S1C appear like they may be switched around. Should the red and blue graphs be switched?"

The graphs are correct as depicted. Note that the stepping assay for our 'minus She1' dataset was performed in very limiting ATP (1 μ M). This leads to a very low stepping rate (i.e., long dwell times) that is equivalent to the stepping rate of dynein in the presence of saturating ATP (1 mM) and 25 nM She1 (brown histogram), but less than the stepping rate of dynein in the presence of only 10 nM She1 (red histogram). **

I'm still confused by these histograms. I understand the slower stepping in lower ATP, but the fit of the dwell times doesn't make sense to me. I don't understand how the dwell times for the blue and brown histograms are smaller than that for the red, given the distribution of values plotted in the graphs. Perhaps I'm missing what k represents here. Is it the time that the motor waits between steps, or the number of steps per sec? In either case I still can't see how the value of k is related to the values in the histograms. I encourage the authors to explain these histograms more fully in the legend/methods.

Detailed Response to Referee #3

1. I still think the use of the term 'mutant' will be misleading for many readers. Most of the time in this field 'mutant' is used to describe a protein with one, or a small set, of changed amino-acids. In this case, the authors are referring to proteins in which entire domains, comprising dozens of amino acids, have been swapped between species.

Per the referee's suggestion, we have changed the term "mutant" where appropriate to clarify our use of the mouse/yeast chimera.

2. "I'm still confused by these histograms. I understand the slower stepping in lower ATP, but the fit of the dwell times doesn't make sense to me. I don't understand how the dwell times for the blue and brown histograms are smaller than that for the red, given the distribution of values plotted in the graphs. Perhaps I'm missing what k represents here. Is it the time that the motor waits between steps, or the number of steps per sec? In either case I still can't see how the value of k is related to the values in the histograms. I encourage the authors to explain these histograms more fully in the legend/methods."

To clarify these data, we have changed the text in the relevant figure legend (Supplementary Fig. 1) to clearly indicate what " k " represents (*i.e.*, the number of steps taken per second), as follows:

"The histograms were fit to a convolution of two exponential functions [$tk^2 \exp(-kt)$] with equal decay constants, k , which reflects the number of steps taken per second^{1,2} ($k \pm$ standard error of the fit is shown)."